# Climate and parameter sensitivity and induced uncertainties in carbon stock projections for European forests (using LPJ-GUESS 4.0)

Johannes Oberpriller[1], Christine Herschlein[2], Peter Anthoni[2], Almut Arneth[2], Andreas Krause[3], Anja Rammig[3], Mats Lindeskog[4], Stefan Olin[4], Florian Hartig[1]

[1] Theoretical Ecology Lab, University of Regensburg, Universitätsstraße 31, 93053 Regensburg, Germany
[2] Department Atmospheric Environmental Research (IMK-IFU), Karlsruhe Institute of Technology, Kreuzeckbahnstr. 19, 82467 Garmisch-Partenkirchen, Germany
[3] Professorship for Land Surface-Atmosphere Interactions, TUM School of Life Sciences Weihenstephan, Technical University of Munich, Freising, Germany
[4] Department of Physical Geography and Ecosystem Science, Lund University, Sweden

Correspondence to: Johannes Oberpriller (johannes.oberpriller@ur.de)

## Abstract

Understanding uncertainties and sensitivities of projected ecosystem dynamics under environmental change is of immense value for research and climate change policy. Here, we analyze sensitivities (change in model outputs per unit change in inputs) and uncertainties (changes in model outputs scaled to uncertainty in inputs) of vegetation dynamics under climate change, projected by a state-of-the-art dynamic vegetation model (LPJ-GUESS v4.0) across European forests (the species *Picea abies*, *Fagus sylvatica* and *Pinus sylvestris*), considering uncertainties of both model parameters and environmental drivers. We find that projected forest carbon fluxes are most sensitive to photosynthesis-, water- and mortality-related parameters, while predictive uncertainties are dominantly induced by environmental drivers and parameters related to water and mortality. The importance of environmental drivers for predictive uncertainty increases with increasing temperature. Moreover, most of the interactions of model inputs (environmental drivers and parameters) are between environmental drivers themselves or between parameters and environmental drivers. In conclusion, our study highlights the importance of environmental drivers not only as contributors to predictive uncertainty in their own right, but also as modifiers of sensitivities and thus uncertainties in other ecosystem processes. Reducing uncertainty in mortality related processes and accounting for environmental influence on processes should therefore be a focus in further model development.

## 1. Introduction

Terrestrial ecosystem models have emerged in the last three decades as a central tool for decision making and basic research on vegetation ecosystems (Cramer et al., 2001; Fisher et al., 2018; IPCC, 2014; Smith et al., 2001; Snell et al., 2014). Projections from different vegetation models, however, often disagree on important details, for example regarding the observable past (Bastos et al., 2020) or the future carbon uptake of forest ecosystems (Huntzinger et al., 2017; Krause et al., 2019). Among the possible reasons for such differences is the uncertainty in climate scenarios (Saraiva et al., 2019), model structural uncertainty (Bugmann et al., 2019; Oberpriller et al., 2021; Prestele et al., 2016), initial condition uncertainty (Dietze, 2017b) as well as uncertainty about the model parametrization (Grimm, 2005), which in turn make models' projections themselves uncertain (Dietze, 2017a). It is widely appreciated that understanding which exact factors drive these uncertainties is of immense value for directing research (Tomlin, 2013), but also to interpret and understand projections (Dietze et al., 2018). For example, the IPCC started in its Fifth Assessment Report to systematically analyze uncertainties and attribute them to model inputs (IPCC, 2014) similar to other predictive sciences (e.g. nuclear reactor safety (Chauliac et al., 2011), energy assessment for buildings (Tian et al., 2018) or policy analysis (Maxim and van der Sluijs, 2011)).

The two main tools to propagate uncertainties in model inputs (drivers, parameters, and model structure) to model outputs are sensitivity analysis (SA) and uncertainty analysis (UA) (Cariboni et al., 2007; Caswell, 2019; Saltelli, 2002; Saltelli et al., 2008). The key difference between these two methods is that an UA considers the magnitude of uncertainty in the model inputs (e.g. parameters, typically determined via expert elicitations and previous studies (Matott et al., 2009)), while a SA is agnostic about the magnitudes of uncertainty in different inputs, and simply calculates the change in the output per unit or percentual change of the respective input (Jørgensen and Bendoricchio, 2001). This difference aside, both methods share the goal of identifying inputs with a high influence on model outputs, with the underlying idea that better constraining these will increase robustness and reliability of model projections (Balaman, 2019).

Although the benefits for understanding model behavior and predictive uncertainties are obvious, relatively few SAs and UAs have been applied to complex ecosystem models and especially the widely used dynamic global vegetation models (DGVMs) that project terrestrial ecosystem responses to climate change or land management (see, e.g., Courbaud et al., 2015; Cui et al., 2019; Huber et al., 2018; Reyer et al., 2016; S. Tian et al., 2014; Wang et al., 2013). A reason for this is arguably the complex structure of most DGVMs (Fer et al., 2018), which makes SAs and UAs computationally demanding and difficult to interpret, especially when performing state-of-the-art global SAs and UAs that compute sensitivities and uncertainties across the entire parameter space (Saltelli et al., 2008) rather than just locally around a reference parameter set (see e.g., Hamby, 1994). Moreover, several studies highlight that sensitivities and uncertainties of DGVMs also exist with respect to environmental drivers (Barman et al., 2014; Wu et al., 2017, 2018), especially solar radiation (Barman et al., 2014; Wu et al., 2018), temperature (Barman et al., 2014) and precipitation (Wu et al., 2017), and it is reasonable to expect that

there can be interactions between parameter and environmental sensitivities, meaning that certain parameters are more
sensitive in some environments than in others. It therefore seems important to investigate parametric sensitivities in
conjunction with their environmental sensitivities in one combined analysis.

In this study, we concentrate on a well-established and widely applied DGVM, the Lund-Potsdam-Jena General Ecosystem
Simulator (LPJ-GUESS) (Gerten et al., 2004; Sitch et al., 2003; B. Smith et al., 2001). Three previous SAs or UAs for the
LPJ family identified the intrinsic quantum efficiency of $CO_2$ uptake (*alpha_C3*) and the photosynthesis scaling parameter
(from leaf to canopy) (*alpha_a*) as the main contributors of sensitivity for net primary production (NPP) (about 50-60% of
the overall sensitivity, Zaehle et al., 2005; Pappas et al., 2013) or foliage projective cover (Jiang et al., 2012). Additionally,
these previous studies show that LPJ-GUESS projections of NPP and vegetation carbon pools showed high sensitivity to tree
structure-related (sapwood to heartwood turnover rate, longevity of trees, Pappas et al., 2013; Wramneby et al., 2008; Zaehle
et al., 2005), establishment-related (maximum sapling establishment rate, minimum forest floor photosynthetically active
radiation for tree establishment, Jiang et al., 2012; Wramneby et al., 2008; Zaehle et al., 2005), mortality-related (threshold
for growth suppression mortality, Pappas et al., 2013) and water-related parameters (minimum canopy conductance not
associated with photosynthesis, maximum daily transpiration, Pappas et al., 2013; Zaehle et al., 2005). Regarding
uncertainties, strong impacts on LPJ-GUESS projections of NPP and vegetation carbon pools (FPC for Jiang et al., 2012)
were found for photosynthesis related parameters (Jiang et al., 2012; Zaehle et al., 2005), but also for water-related
(minimum canopy conductance not associated with photosynthesis, Zaehle et al., 2005) as well as structure-related
parameters (tree leaf to sapwood area ratio, crown area to height function Jiang et al., 2012), whereas soil hydrology
parameters were not identified as very sensitive in earlier studies.

Since the publication of these studies, however, the structure of the LPJ-GUESS model changed substantially. The most
important changes are the inclusion of the nitrogen cycle (Smith et al., 2014) and new management modules (Lindeskog et
al., 2021). Since these changes, no study has systematically examined how model sensitivities and uncertainties were
affected by the new model structure. Moreover, previous SAs and UAs ignored management parameters, which, however,
are expected to have large impacts on carbon pools and fluxes (Lindeskog et al., 2021).

A further limitation of most previous studies for LPJ-GUESS and other models (e.g. Mäkelä et al., 2020) is that they either
analyzed sensitivities and uncertainties to parameter changes, or to changes in the environmental drivers, but not both. As
discussed earlier, however, there are good reasons to expect that the sensitivity of parameters will change if environmental
drivers change. Given that previous sensitivity analyses used different choices for these boundary conditions (different
sensitivities for the climate scenarios and sites in Jiang et al., 2012; for different elevations in Pappas et al., 2013; different
sites in Wramneby et al., 2008), this not only limits the comparability between studies, but also questions the generality of
the results for all climatic conditions. Only Jiang et al. (2012) combined parameter and driver sensitivities, but used for the

latter only a number of fixed climate scenarios instead of a range of possible values, which prohibits a systematic joint analysis. Moreover, it would be interesting to compare the relative importance of drivers and parameters for the predictive uncertainty of model simulations and how these change between environmental zones (here we use the classification of Metzger et al., 2005) and thus on an environmental gradient. When sensitivities or uncertainties of parameters belonging to a specific process increase on an environmental gradient, this indicates that the process itself becomes more important on the gradient (Saltelli, 2002). By comparing such changes to existing ecological hypotheses, we can test if model sensitivities and thus process descriptions are in line with ecological expectations.

To answer these questions, we analyzed sensitivities and uncertainties in LPJ-GUESS for 200 randomly distributed sites across Europe (see Appendix A1.1). We address the issue of interactions between environmental and parametric sensitivities by simultaneously investigating uncertainty in environmental drivers (precipitation, temperature, solar radiation, $CO_2$, nitrogen deposition) with parametric uncertainty in the most important processes (photosynthesis, establishment, nitrogen, water cycle, mortality, disturbance/management, and growth) for dynamic climate change from 2001-2100 and steady climate from 2100-2200. We simulated the most abundant tree species in Europe (*Fagus sylvatica, Pinus sylvestris* and *Picea abies*) individually and in mixed stands, as these species are suffering from climate change (e.g. Buras et al., 2018; Walentowski et al., 2017) and could benefit from mixed stands (e.g. Pretzsch et al., 2015). To test climate change impacts, we randomly sampled climate projections within the boundaries of RCP2.6 and RCP8.5. Thereby, our key objectives were to understand the sensitivities and uncertainties of LPJ-GUESS due to environmental drivers and parameters. We were especially interested in 1) overall sensitivities and uncertainties across European forests, 2) uncertainties per environmental zone and 3) uncertainties on a temperature gradient. Moreover, we investigated, 4) if and how environmental conditions change the uncertainties of environmental processes.

## 2. Methods and Material

### 2.1. The LPJ-GUESS vegetation model

LPJ-GUESS is a process-based ecosystem model that simulates vegetation growth, vegetation dynamics and biogeography as well as biogeochemical (e.g. nitrogen and carbon) and water cycles (Lindeskog et al., 2013; Olin et al., 2015; Smith et al., 2014). Ecosystem dynamic processes in the model include establishment, growth, mortality, and competition for light, space and soil resources. To simulate these processes, the model combines time steps on different scales from daily (e.g. phenological and photosynthesis processes) to yearly (e.g. allocation of net primary production to tree carbon components) basis. LPJ-GUESS includes forest gap dynamics succession of cohorts (each represented by an average individual) of different plant functional types (PFTs) or species. Each PFT/species has a unique parameter set.

In this study, we use a model version that was slightly modified from Lindeskog et al. (2021), which is based on the LPJ-
GUESS 4.0 version, with a re-parameterization for spruce (*Picea abies*), pine (*Pinus sylvestris*) and beech (*Fagus sylvatica*)
(see Appendix A1.2 for *Pin. syl.* and *Pic. abi.*). To account for the stochastic components of establishment, mortality and
patch destroying disturbances, LPJ-GUESS simulates several replicate patches (25 for the simulation with the reference
parametrization and 1 for each simulation in the SA and UA) representing "snapshots" of the grid-cell. In this model version,
fire is based on the BLAZE model (Rabin et al., 2017). Thereby annually burned area is generated based on fire weather and
fuel continuity and distributed to monthly intervals based on climatology (Giglio et al., 2010). Tree mortality is then
estimated by computing firelines based on weather and converted into height-dependent survival probabilities (see Haverd et
al., 2014) depending on empirical biome specific parameters.

A first set of key parameters from our expert elicitation (see below) for **establishment** are the bioclimatic limits (i.e.
minimum growing degree days (*gdd5min_est*), minimum 20-year coldest month (*tcmin_est*), maximum 20-year coldest
month (*tcmax_est*) and minimum forest photoactive radiation at forest floor (*parff_min*)), which build the environmental
envelope for establishment. Given the bioclimatic limits are fulfilled, at regular intervals new PFTs are established (here: 1
year) given enough space, light, soil water and photoactive radiation at forest floor is available for establishment (B. Smith et
al., 2001). Moreover, each of our three investigated species has a maximum establishment rate (*est_max*) (B. Smith et al.,

144 2001).


**Structure of trees** in the model is mainly linked to the simulated growth of trees, which is triggered by allocating all net
primary production (NPP) besides a reproduction debt of 10% (*reprfrac*) to tree components thereby satisfying mechanical
(e.g. allometric eq. for the relationship between height and diameter with allometric parameters (*k_allom2, k_allom3*) *(e.g.*
*Huang et al., 1992)*, the relationship between tree leaf to sapwood area (*k_latosa*) (e.g. Robichaud & Methven, 1992), the
relationship between crown area and height (*k_rp*) (packing constraint, Zeide, 1993), the maximum crown area
(*crownarea_max*) and leaf longevity (*leaflong*)) and functional balance as well as demographic constraints (Sitch et al.,
2003). Each living tissue is assigned a turnover rate transferring sapwood into heartwood (*turnover_sap*) and leaves
(*turnover_leaf*) and fine roots (turnover_root) to litter. Investment into above and belowground growth is influenced by the
resource stress as individuals are competing for light, space, nitrogen and water. Competition for light is determined by the
photosynthetic response and light extinction in the canopy. Competition for space (*self-thinning*) is represented in the model
via allometric equations between crown area and stem diameter (Sitch et al., 2003). Competition for nitrogen and water is
determined by tree individual demand for nitrogen and water and soil availability of nitrogen and water and the PFT-specific
root profile. Competition between species will favor certain life-history strategies in particular situations, for example shade-
tolerant (e.g. *Fagus sylvatica* and *Picea abies*) or intermediate-shade tolerant (e.g. *Pinus sylvestris*) growth responses, and
dynamically changing root-to-shoot ratios.

**Tree mortality (natural or via harvest)** in the model responds to growth efficiency (ratio of annual NPP to leaf area) being
too low over a 5-year period, e.g. due to light competition, maximum longevity of a PFT or changes in environmental
conditions (e.g. tolerance to drought (*drought_tolerance*) changes water uptake) exceeding the species suitable range. Light
competition is modeled using the foliage projective cover (FPC), defined as the area of ground by foliage directly above it,
using Beer's Law (B. Smith et al., 2011). The resulting shading mortality is distributed proportional to species' FPC growth
in the respective year due to their biomass increase. Mortality is modeled inversely proportional to the growth efficiency
(with a given species-specific threshold (*greff_min*), e.g. Waring (1983)). Moreover, negative NPP of a species kills all
individuals of the respective cohort. Background mortality probability increases with tree age, reaching one at the maximum
longevity (*longevity*). Mortality has also a stochastic component. Natural disturbances are implemented in the model as
process-based wildfires (with a given fire resistance for each species (*fireresist*)) and as patch-destroying disturbances (e.g.
windthrow and landslides) with the same yearly occurrence probability for all patches (inverse of *distinterval*). Additional
mortality arises from forest management activities, determined by thinning intensity (percentage of all trees cut,
*thinning_intensity*) and cutting intervals (*cut_interval*), which can be set for each species individually. For a more detailed
description of the management module and the additional management parameters see Lindeskog et al. (2021).

**Nitrogen** input is implemented in the model through nitrogen deposition (prescribed) and biological nitrogen fixation. The
latter is simulated empirically as a linear function with intercept (*nfix_a*) and slope (*nfix_b*) of the five-year averaged actual
evapotranspiration (Cleveland et al., 1999). The resulting amount of nitrogen accumulates in the ecosystem equally over the
year and directly adds to the available mineral soil nitrogen pool. When nitrogen is in living tissue, a fraction (*nrelocfrac*) is
re-translocated before leaf- and root shedding.

**Photosynthesis** is modeled as a function of absorbed photosynthetically active radiation, temperature (optimum temperature
range for photosynthesis determined by *pstemp_low and pstemp_high*, Larcher, 1983), intercellular $CO_2$ (i.e. non-water
stressed ratio of intercellular to ambient $CO_2$ (*lambda_max*)), and canopy conductance thereby considering a species-specific
respiration coefficient (*respcoeff*) (B. Smith et al., 2001) and nitrogen availability. The photosynthesis scheme is a modified
version of the Farquhar photosynthesis model, but instead of prescribed values for the Rubisco capacity it is optimized for
maximum net $CO_2$ assimilation at the canopy level (Smith et al., 2014).

**Water** availability for plants is based on precipitation and snowmelt in the two-layer soil hydrology submodule (for details
see Hickler et al., 2004; Smith et al., 2001). Vegetation transpiration and evaporation (with a maximum evapotranspiration
rate (*emax*)) from bare ground and leaves reduce water availability as well as runoff from saturated soil (Sitch et al., 2003).
Water vapor exchange by the vegetation canopy is calculated on a daily basis within the photosynthesis scheme (e.g.
minimum canopy conductance not associated with photosynthesis (*gmin*)). The water supply and transpirative demand are
calculated on a daily basis and converted into a drought-stress coefficient. Given this coefficient, the investment in roots at
the costs of leaves is calculated.

## 2.2. Simulation setup

We selected 200 study sites (see Appendix A1.1) spatially and environmentally stratified over Europe by applying random
stratified sampling (using the R package splitstackshape Mahto, 2019) with longitudinal and latitudinal coordinates as well
as mean precipitation, solar radiation and temperature as categories based on IPSL-CM5 Earth System Model CMIP5
(Dufresne et al., 2013) climate data. We chose 200 sites as a compromise between the high computational demand of
running LPJ-GUESS multiple times for all sites and a good spatial as well as environmental coverage of Europe. For these
sites, we performed simulations for each of the three most common species in Europe (*Fagus sylvatica*, *Pinus sylvestris* and
*Picea abies)* as monospecific stands and additionally all three species together as mixed stands.

The simulation period was from 1861 to 2199. To start the simulations with equilibrium C pools and fluxes, we spun up LPJ-
GUESS vegetation and soil carbon and nitrogen pools to pre-industrial equilibrium by recycling the 1861 to 1900 climate,
the 1861 $CO_2$ concentration (Meinshausen et al., 2011) and nitrogen deposition. For the transient and future simulation runs,
we used the bias-corrected monthly IPSL-CM5 Earth System Model CMIP5 (Dufresne et al., 2013). From this data set, we
extracted temperature, precipitation, number of wet days per month, and incoming solar radiation from 1861 to 2099 for
RCP4.5 as base scenario and RCP2.6/RCP8.5 as lower/upper boundaries for the climate ranges (see below). In addition to
these data, monthly nitrogen deposition was extracted from Lamarque et al. (2013) and soil texture data from Batjes (2005).
All these driving data had a spatial resolution of 0.5°x 0.5°. We recycled detrended data from 2090-2099 for all
environmental drivers except $CO_2$ and nitrogen deposition and used these as potential stable climates for the 2100-2199
period.

## 2.3. Selection of parameters and drivers and their ranges

The a priori selection of the most influential parameters that can be specified in the parameter file and their ranges was based
on our expert knowledge (following the SHELF expert elicitation protocol, see Gosling, 2018) and a literature review. The
resulting eleven (= 33%) parameters common for all species and 22 (= 20%) species-specific parameters (see Table 1) were
grouped to the specific processes they contribute most to (Table 1, Grouping).

From the environmental drivers of the model, we selected incoming solar radiation, temperature, precipitation, atmospheric
$CO_2$ and nitrogen deposition for our analysis. To obtain uncertainties for temperature, precipitation and solar radiation, we
calculated the mean deviations of RCP8.5/RCP2.6 to our base scenario RCP4.5 plus/minus one standard deviation as
maximal/minimal per site. As the $CO_2$ data is global and not site-specific, we calculated ranges from the global data set
(RCP2.6 as minimum, RCP8.5 as maximum) averaged over time and plus/minus a standard deviation. For nitrogen
deposition, we used RCP6.0 as maximum and RCP2.6 as minimum with the same procedure as for the other drivers.

**2.5. Sensitivity analysis and uncertainty analysis**
LPJ-GUESS predicts a substantial number of output variables, which could all be examined regarding their sensitivities and
uncertainties. Here, we concentrate on carbon outputs (**gross primary production GPP**, **total standing biomass TSB** and
**net biome productivity NBP**), because of forests' role for carbon cycling (Bonan, 2008), their large contribution to the land
carbon sink (Pugh et al., 2019) and the economic importance of tree growth for forest owners (Pearce, 2001).

Sensitivities and uncertainties were calculated by Monte-Carlo sampling from the assumed multivariate parameter and
climate uncertainty. For the monospecific / mixed simulations, we drew respectively 10.000 / 50.000 parameter and climate
combinations randomly from the prespecified uncertainty ranges, and ran the model based on these combinations for each of
the 200 sites. Note, that for mixed simulations, for each simulation we individually drew parameter combinations for each
species, i.e. the same parameter could be different for different species. In total, this means that 200 x (50.000 + 3 x 10.000)
= 16 million LPJ-GUESS simulations were run.

We quantified sensitivity and uncertainty indices by running multiple linear regressions with the model output averaged over
time as response, and parameters and drivers as well as their second order interactions as predictors. With 200 sites, each
having three monospecific and one mixed stands setup, we overall ran 200x (3 +1) = 800 linear regressions. This analysis
corresponds to a global SA/UA in the context of regression analysis and has been applied to other system models (e.g. Sobie,
2009). The estimated effects from the regression can be interpreted as sensitivities, as the effect of a unit change of the driver
on the response (model output) is estimated. By scaling the predictors to the range [-0.5, 0.5], we obtained the corresponding
uncertainties. To check whether we missed non-linear effects, we additionally applied a random forest and extracted the
variable importance (following Augustynczik et al., 2017, see Appendix A1.3.). To calculate mean sensitivities/uncertainties
for each species, we averaged site-specific sensitivities over all sites with an average annual biomass production greater than
2 tC/ha. We have chosen this threshold because smaller values indicate that the environment is not suitable for the species,
however, for each site at least one species was able to establish. For the mixed stands, we first averaged the three species-
specific sensitivities/uncertainties per site and then averaged over all sites. Mean percentual sensitivities were calculated by
dividing by the mean model output, while mean uncertainty contributions were calculated by dividing by the entire
uncertainty budget. Thereby, positive values mean that the respective output increases with increasing parameter values,
while negative values mean that it decreases.

It is important to note that uncertainties and sensitivities have different interpretations, and which of these two is more
relevant strongly depends on the purpose. The calculated percental sensitivities can be interpreted as percentage change in
the corresponding output, when changing a parameter value 1% in the prespecified range. The calculated uncertainties per
parameter/driver can be interpreted as relative proportion of the overall uncertainty budget coming from environmental
drivers and parameters. For scenario-analysis, e.g. comparing different cut intervals of forests, sensitivities provide a direct
estimate of the model response, e.g. how much biomass changes when the cut interval is changed. For a comparison of
different model forecasts, uncertainties are usually more relevant. If a reduction of uncertainty via a model-data comparison
is the purpose, both measures are important, as parameters with high sensitivities can contribute more or less predictive
uncertainty, depending on their input uncertainty.
**3. Results**
**3.1. Mean sensitivities over Europe**
Regardless of the output variable, LPJ-GUESS was most sensitive to photosynthesis-related parameters (*respcoeff,*
*lambda_max*), parameters controlling the wood turnover (*turnover_sap*) and tree allometry (*k_rp*), water-related parameters
(*emax*), mortality-related parameters (*greffmin*) and environmental drivers (temperature, $CO_2$ and solar radiation) (Fig. 1).
When looking at differences in the strength of sensitivities for different outputs, TSB was most sensitive to the respiration
coefficient (*respcoeff*), the growth suppression mortality threshold (*greff_min*) and solar radiation while NBP projections
showed negative sensitivity to wood turnover rates (*turnover_sap*) and *longevity* and positive sensitivity to temperature, $CO_2$
and the ratio of intercellular to ambient $CO_2$ (*lambda_max*). GPP was negatively sensitive to the respiration coefficient
(*respcoeff*), growth suppression mortality threshold (*greffmin)*, tree allometry (*k_rp)* and temperature and positive to $CO_2$,
solar radiation and the maximum transpiration rate (*emax*). Establishment and nitrogen showed the smallest sensitivities for
all three carbon-related projections (Fig.1). Note also that NBP had higher percental sensitivities than GPP and TSB.

Mixed stands were less sensitive to changes in parameters than mono-specific stands (Fig. 1). For monospecific simulations,
species sometimes showed different magnitudes and even directions of sensitivities, especially *Fag. syl.* was more strongly
affected by bioclimatic limits and *Pin. syl.* showed higher sensitivity to environmental drivers (temperature and solar
radiation) than the other species. Moreover, TSB and GPP are negatively sensitive to temperature except for Fag. syl. For
NBP, the direction of sensitivities changes between species for the non-water-stressed ratio of intercellular to ambient $CO_2$
*(lambdamax)*, the respiration coefficient (*respcoeff*), the root turnover (*turnoverroot)*, an allometric constant (*krp)* and the
maximum evapotranspiration rate (*emax)*.


## 3.2. Mean uncertainties over Europe

Looking at uncertainties, we found that environmental drivers contributed most of all processes/drivers to the predictive uncertainty (Fig 2), regardless of the considered model output. For TSB projections, $CO_2$, solar radiation and temperature contributed substantial uncertainty (Fig. 2a). Additionally, large uncertainty contributions arose from growth suppression mortality thresholds (*greffmin*) and the respiration coefficient (*lambda_max*). Uncertainty in NBP projections was substantially affected by model parameters (*longevity* (Mortality process), *tcmax_est* (Establishment process), *turnover_sap* (Tree structure process), *greffmin* (Mortality process) and *emax* (Water process)), additionally to the high contributions of temperature and $CO_2$ (Fig. 2b). For GPP projections, solar radiation and $CO_2$ contributed most to climate induced uncertainty, while the threshold for growth suppression mortality (*greffmin)* and maximum evaporation rate (*emax*) contributed most to parameter induced uncertainty (Fig. 2c). Notably, also nitrogen-fixation induced uncertainty was substantial (7-9%) for TSB and GPP. Most tree structure related parameters except the sapwood to heartwood turnover rate (*turnoversap)* and the fraction of NPP allocated to reproduction (*repfrac*) contributed only small uncertainties (Fig. 2). Uncertainty contributions analyzed by a random forest are similar to linear regression results (see Appendix 1.3.).

By analyzing uncertainty contributions on a species level, a more diverse picture emerged. *Fag. syl.* was more affected by temperature and less by solar radiation than the other species. Additionally, we found that uncertainty contributions of environmental drivers were substantially higher for mixed than for mono-specific stands.

## 3.3. Geographic variation in uncertainties of TSB across Europe

To project the uncertainties of TSB (for GPP and NBP see Appendix 1.4.) into the European environmental space, we filtered stands according to environmental zones, then calculated mean uncertainties per environmental zone and aggregated these per process.

The broad pattern of TSB uncertainty contributions for all three monospecific and mixed stands remains similar in all environmental zones. On average across all environmental zones, stands and species about 45% of the uncertainty was due to environmental drivers, 15% due to mortality-, 14% due to photosynthesis-, 12% due to structure-, 7% due to water- and 7% due to nitrogen-related parameters (Fig. 3).

For the individual environmental zones, however, there were subtle differences. In the Mediterranean mountain (MDN) and Pannonian (PAN) zone, environmental driver induced uncertainty was higher than on average especially for monospecific stands (Fig. 3). In the Boreal (BOR), Atlantic central (ATC), and Atlantic north (ATN) zone, tree structure- related

uncertainty increased compared to the average pattern (Fig. 3). In the Atlantic central (ATC) and Atlantic north (ATN) zones
nitrogen related uncertainty increased for all species and stands (Fig. 3).

To examine this spatial pattern further, we investigated the change of uncertainties across a temperature gradient. To this
end, we aggregated the uncertainties per site and process/driver and then fitted a linear regression with the process/driver as
predictor and the aggregated uncertainties as dependent variables.

For TSB, we found that increasing mean annual temperature increased the uncertainty contributions of environmental
drivers, water- and establishment-parameters, while the uncertainty due to nitrogen- and tree structure- related parameters
decreased (Fig. 4a). Thereby, the uncertainty contributions of environmental drivers ($\approx 0.4\%/°C$) increased the most
(measured in percentage points per $°C$) and uncertainty contributions of nitrogen fixation decreased most ($\approx -0.5\%/°C$).
Mortality and photosynthesis stayed approximately constant on the gradient (Fig. 4b).

Looking in more detail at the environmental drivers, temperature ($\approx +0.75\%/°C$) as well as $CO_2$ ($\approx +0.2\%/°C$) and
precipitation ($\approx +0.25\%/°C$) induced uncertainty increased with mean annual temperature, while the uncertainty contribution
of solar radiation ($\approx -0.75\%/°C$) decreased with mean annual temperature (Fig. 4c). Nitrogen deposition induced uncertainty
contributions stayed approximately constant on a mean annual temperature gradient.

The above geographical and correlative observations of changing uncertainties across Europe receive further support when
looking at the interactions between uncertainties of different drivers/parameters (Fig. 5). Interaction indices were calculated
by averaging the interactions found in the linear regression over all sites and species (Fig. 5b). Moreover, to investigate the
overall influence on other parameters or drivers we summed the absolute individual interaction indices of each parameter
with each other (Fig. 5a).

We found that environmental drivers (temperature, solar radiation, $CO_2$ and precipitation) had the highest sum of interactions
for TSB (Fig. 5a). Moreover, the respiration coefficient (*respcoeff*), the growth suppression mortality threshold (*greffmin*),
longevity, the sapwood to heartwood turnover rate (*turnover_sap*) and maximum evaporation rate (*emax*) had a lower, but
still high sum of interactions (Fig. 5a). Establishment and nitrogen related parameters had only a few weak interactions (Fig.
5). Strong interaction effects occurred mostly with environmental drivers (Fig. 5b). A main part of these interactions was
between the different environmental drivers themselves (solar radiation- $CO_2$ and solar radiation- $CO_2$). Additionally, we
found interactions of parameters and environmental drivers (temperature-sapwood to hardwood turnover (*turnover_sap*),
temperature – threshold for growth suppression mortality (*greffmin)* and temperature-respiration coefficient (*respcoeff)* (Fig.
5b)) and moderate parameter-parameter interactions (*longevity* (Mortality process) - *greffmin* (Mortality process), *respcoeff*
(Water process) – *longevity* (Mortality process) (Fig. 5b)). Similar patterns were present for the other two carbon outputs
(see Appendix A1.4.).

## 4. Discussion

In this study, we analyzed sensitivities and uncertainties of the LPJ-GUESS vegetation model due to environmental driver
and parameter variations across European forests. We found that the model is most sensitive to relative (percentage) changes
in photosynthesis-related parameters, structure-related parameters controlling the wood turnover and tree allometry, water-
related parameters, mortality-related parameters, and environmental drivers (Fig.1), irrespective of the considered output
variable. When considering the different uncertainties (i.e. the entire plausible range) in these parameters and the
environmental inputs, we found that environmental drivers and parameters controlling evapotranspiration, background
mortality and nitrogen cycling contribute most to predictive uncertainty (Fig. 2). When correlated against a temperature
gradient and thus geographically from north to south, uncertainty contributions to TSB increased for environmental drivers
and decreased for tree structure and nitrogen-related parameters (Fig. 3, 4). Interactions between the uncertainty
contributions were mainly between different drivers or between model parameters and drivers, whereas only a few
parameter-parameter interactions were present (Fig. 5).

Our finding that average sensitivities of carbon-related projections across European forests were highest for photosynthesis-
related parameters amplifies the evidence from earlier studies (Pappas et al., 2013; Zaehle et al., 2005), although we have
used different parameter ranges. In addition, the finding about high sensitivity of LPJ-GUESS to parameters controlling tree
structure and especially carbon turnover (*turnover_sap*) (Fig. 1) is in line with results reported for a previous version of LPJ-
GUESS (Pappas et al., 2013) and its important role for carbon allocation in trees found in empirical studies (e.g. Herrero de
Aza et al., 2011). The finding that carbon-related projections are very sensitive to mortality-related parameters (*greffmin*) is
also supported by previous studies on the sensitivity of vegetation models and underlines the importance of improving
mortality submodules for generating precise projections of vegetation dynamics (Bugmann et al., 2019; Hardiman et al.,
2011). Moreover, sensitivities in mixed stands were lower than in mono-specific stands for NBP and GPP (Fig. 1) (in line
Wramneby et al., 2008). The reason for that imbalance may be that other species can dampen and even benefit from non-
optimal life-history strategies of an individual species (Loehle, 2000). Another reason might be, that for mixed simulations
we sampled parameters for each species individually, which reduces the influence of each parameter on stand-level carbon
projections.

We found that uncertainty contributions of environmental drivers were comparable to the uncertainty contributions of all
parameters together (Figs. 2-5, see also Snell et al., 2018 for the FLMs model, but see Petter et al., 2020, who found that
most uncertainty is induced by the choice of the forest model). Especially high uncertainty contributions arose from

temperature (negative effect for TSB, GPP positive for NBP), $CO_2$ (positive effect for all variables) and solar radiation (positive effect for all variables). These results are supported by the earlier studies on the effect of environmental drivers in DGVMs (Barman et al., 2014; Wu et al., 2017, 2018). The positive effect of $CO_2$ could be explained by increased water-use efficiency and the $CO_2$ fertilization effect (also found for other DGVMs Keenan et al., 2011; Galbraith et al., 2010), which in LPJ-GUESS is an emerging property of the formulation of photosynthesis and respiration (see Hickler et al., 2008). However, empirical studies do not find such an effect (Körner, 2006), which could be link to the fact that LPJ-GUESS does not model phosphor cycling which could be the limiting nutrient (for a DVGM study see Fleischer et al., 2019). We speculate that the negative effect of temperature (also found for multiple DGVMs, see Galbraith et al., 2010) arises from decreased photosynthetic efficiency and increased respiration rates with higher temperatures (see the empirical study of Gustafson et al., 2018, here confirmed by the negative relationship between temperature and the respiration coefficient). This effect, however, differed in magnitude and direction between tree species (Fig. 2) - while there was a strong effect for *Pic. abi.* and *Pin. syl.*, *Fag. syl.* was less affected, which could be a sign of its higher resistance to increasing drought (Buras and Menzel, 2019; Tegel et al., 2014; but see Charru et al., 2010). From the parameters, especially water-, nitrogen- and mortality-related parameters contributed a substantial amount of uncertainty. The uncertainty contributions from mortality parameters (Bugmann et al., 2019, for a variety of DGVMs) and water (Pappas et al., 2013, with different parameter ranges for LPJ-GUESS) were already highlighted by earlier studies.

## 4.1. Geographical and environmental patterns in sensitivities and uncertainties

Several of our results suggest that environmental context influences the sensitivity of LPJ-GUESS model parameters. First, we found changing uncertainties across different vegetation zones (Fig. 3) and on an environmental gradient (Fig. 4) and that most interactions occurred with environmental drivers (Fig. 5). Moreover, uncertainty contributions analyzed by a random forest were similar to the linear regression results, but assign higher importance to environmental drivers (see Appendix A1.3). All these findings indicate that environmental context can change the importance of different processes in the model, which is in line with the biological expectation that the environment affects the physiology of organisms directly and thus indirectly the fitness and biotic interactions (e.g. Seebacher & Franklin, 2012; Tylianakis et al., 2008), and that environmental responses can be particularly nonlinear (e.g. Burkett et al., 2005) or show higher order interactions.

Interestingly, our results of decreased uncertainty contributions of structure- related parameters and increased contributions of environmental drivers on the temperature gradient (Fig. 4) also seem in line with the stress-gradient hypothesis (Maestre et al., 2009), an empirically-observed pattern which states that in stressful environments, positive interactions should occur more often than in benign environments (e.g. Callaway, 2007). For the ecosystem that we consider, we interpret increasing temperature as increasing stress (e.g. Ruiz-Pérez and Vico, 2020), and structure as the best indicator for competitive interactions as the structure dictates resource allocation (e.g. bigger crown, but identical stem diameter leads to more

photosynthesis; more sapwood to heartwood turnover requires less NPP). With this interpretation, one would conclude that
under increasing stress, the importance of competition-related parameters decreases in the model, as expected from the
stress-gradient hypothesis. We acknowledge that a fair amount of interpretation is needed to arrive at this conclusion, and we
do not claim that this result lends evidence to the empirical discussion about the generality of the stress-gradient hypothesis,
but we find it noteworthy that such a large-scale pattern emerges in the model from lower-level processes, without having
been imposed (see also Levin, 1992).

**4.2. Associated uncertainties of previous changes in model structure and implications for future model development**

The management and the nitrogen cycling module are the most recent improvements of the LPJ-GUESS model (Smith et al.,
2014; Lindeskog et al., 2021). Compared to previous sensitivity and uncertainty analysis, the high contributions of the
nitrogen fixation to the predictive uncertainty of TSB and GPP (Fig. 2 a,c) are novel, though not surprising, as nitrogen is an
important factor for the productivity of most temperate and boreal ecosystems (Vitousek and Howarth, 1991). The main
reason why few earlier studies report those uncertainties is that vegetation models have only recently begun to integrate
nitrogen cycling and limitation (e.g. B. Smith et al., 2014). The management module showed only small uncertainties, which
could be due to the narrow parameter ranges for the cut interval and thinning intensity reflecting typical forest owners'
choices. As forest owners usually try to maximize their profits (Johansson, 1986; but see Brazee and Amacher, 2000) and
thus biomass production, low sensitivities of the management module are not surprising. A more suitable and important test
case and application of the management module is a historical reconstruction of foliage projective cover data or similar
outputs of the LPJ-GUESS model.

Our study helps to guide the model application, discussion of uncertainties and model development of LPJ-GUESS and other
DGVMs. First, future model applications and model comparisons should focus on mortality as these processes contributes
high uncertainties for carbon-related projections (see Fig. 1-3). Thereby, it should be investigated if these uncertainties stem
from the intra-specific variability of the parameters itself (Bolnick et al., 2011), parameters are just not identifiable (see
Marsili-Libelli et al., 2014), or if a model data comparison could reduce uncertainties in the parameters (e.g. Hartig et al.,
2011). Using time series inventory data might help as it is informative for constraining mortality modules (Cailleret et al.,
2020). Second, small sensitivities of establishment related parameters are surprising as we know that not all three
investigated species can effortlessly establish across all of Europe, e.g. Fag. syl. can only establish on locations with no
extreme drought and heat and no extreme winter frosts (Bolte et al., 2007). Thus, either we missed important parameters of
this module, or the parametrization of the model needs to be updated. Third, when introducing new processes or coupling
with other models (e.g. Forrest et al., 2020) calculating interactions helps to get a first impression where these new processes
influence other model processes and potentially detect missing links. Moreover, future model applications can interpret their
results with regard to the sensitivities in different factors (Saltelli et al., 2019) and discuss uncertainties and the causing
factors, when used in policy advice (Laberge, 2013).

**4.3. Limitations**

We caution that our results regarding the importance of different factors for predictive uncertainties (but not sensitivities)
depend on the a priori defined uncertainty range of the contributing factors (see Wallach & Genard, 1998), as well as on
several other technical choices in our study. For determining uncertainty ranges of the drivers, we used RCP scenarios;
however, these were not created as probabilistic min / max ranges. For the model parameters, we relied on expert guesses,
reducing subjectivity as far as possible by following the SHELF expert elicitation protocol (Gosling, 2018). Future studies
could include more experts and their opinion on parameter distributions to reduce variability in this protocol. As the model is
sensitive to parameters and environmental drivers, and because these influence each other, we treated them in a combined
sensitivity and uncertainty analysis (Saltelli et al., 2019), however, when interpreting it should be kept in mind that the one
group relates to uncertainties in the model, while the other is external, so the two are conceptually very different. A certain
ambiguity also arises from the definition of the indicators: here, we calculated sensitivities and uncertainties by capturing
only linear components and second-order interactions, and we may therefore miss highly non-linear (and in particular hump-
shaped) responses in LPJ-GUESS (Roux et al., 2021). However, our comparison to uncertainties calculated with random
forest variable importance, a method that would also capture nonlinearities, did not reveal any qualitative differences in the
ranking of parameter importance (Appendix A1.3). Overall, while we acknowledge that a certain amount of subjectivity
exists in the choice of input uncertainty and calculation of indices, we believe that our results are quantitatively robust to
those choices.

Moreover, we acknowledge that LPJ-GUESS is known to be sensitive to the scaling parameters *alpha_a* and *alpha_C3*
(Pappas et al., 2013; Zaehle et al., 2005), which we have omitted from our analysis. These parameters, however, are not
accessible in the parameter input file. Instead, they are hard coded in the model's source code and therefore a normal user
would not change them. We argue that these parameters should thus be counted towards the more general and here neglected
contribution of structural uncertainty (i.e. the uncertainty regarding the functional form of processes or even to entire
modules) to the joint model uncertainty. Several previous studies suggest that the sensitivity of vegetation models to
structural changes can be large, often larger than to parameters (e.g. Bugmann et al., 2019), and it would certainly be useful
(although very complicated) to explore these uncertainties together with the here considered factors in a joint analysis. In the
present study, however, we considered only the parameters that would be accessible to normal LPJ-GUESS users, and
neglect structural uncertainty that could be explored by changing the source code.

## 5. Conclusions

Our findings highlight the relative importance of parametric uncertainties in different processes and their interactions with uncertainties in environmental drivers for carbon projections with LPJ-GUESS. Our results demonstrate that environmental context changes uncertainty contributions of other processes across the European environmental gradient. The pattern of decreasing importance of competition towards the warmer areas is in line with the stress-gradient hypothesis, which posits that the importance of competition decreases with increasing environmental stress. Our findings improve our understanding of forest ecosystem models, enable pathways for future ecosystem model development and thus builds a basis for more realistic projections. In the future, parametric uncertainties could be reduced by model-data fusion (e.g. Trotsiuk et al., 2020) of LPJ-GUESS, concentrating on the parameters contributing most uncertainty in each geographic region (Fig. 3). Reducing uncertainties in the drivers is more difficult. To some extent, environmental drivers are themselves influenced by the vegetation (Strengers et al., 2010), so model-data fusion on a fully coupled model including feedback loops between vegetation and climate, as well as a general improvement of climate models, could reduce driver uncertainty to some degree. Effectively, however, much of the uncertainty in this section arises from potential greenhouse gas emission trajectories, for which a probabilistic assignment is difficult due to their dependency on human decision-making.

**Appendix A**

**A1.1 Site selection**

We sampled 200 sites geographically and environmentally stratified over Europe and thereby avoided sites near the sea. The corresponding sites with the average temperature (Fig. A1) covers most of European climates and vegetation zones.

**A1.2. Re-parametrization for better fit to observed data**

There are several technical and methodological reasons requiring a re-parametrization of LPJ-GUESS for our study. First, most of European forests are managed and species are planted far outside of their natural distribution. Second, the introduction of the nitrogen cycle (Smith et al., 2014) changed the model structure and thus parameters require an adjustment. Third, the productivity of trees in managed forests did not fit to the reported inventory data (Fig. A2). To account for all these issues, we adjusted the parametrization of (Hickler et al., 2012) to allow species growing according to their actual (i.e., caused by forest management) distribution instead of their natural distribution.

Especially *Picea abies* and *Pinus sylvestris* are planted far outside their natural distribution (Figure S2). In particular we
adjusted bioclimatic limits, drought tolerances, longevity, leaf turnover, disturbance intervals and allometry for these species.
**A1.3. Random forest results**
To check the consistency of the results obtained via linear regressions, we compare them to variable importance of random
forest. The variable importance measures additionally non-linear effects and thus, should be able to deal with non-linear
models like DGVMs. We calculated the variable importance the same way as we did for the linear regression by fitting a
random forest with all parameters against the sum of differences between model outputs with default values and model
outputs with sampled parameters. As our parameters were sampled from a uniform distribution with no correlation between
the individual parameters, random forest variable importance can be compared to linear regression results.

The ranking is very similar to the ranking of the parameters and environmental drivers obtained via linear regression (Fig.
A3). There is, however, a difference in the magnitude of the uncertainty induced by drivers, which is higher compared to
linear regression (Fig A3). The higher uncertainty due to drivers is thus a nonlinear effect and stresses our conclusion that
environmental conditions change the uncertainty contributions of other parameters.

**A1.4. Interactions of GPP and NBP**
Interactions of gross primary production (Fig. A4 a,b) and net biome production (Fig. A4c,d) are similar to the interactions
of total standing biomass. These interactions are mostly between environmental drivers and environmental drivers or
between environmental drivers and parameters (Fig. A4). Some strong interactions are between parameters and parameters,
however, in such interactions there are always parameters included having strong interactions with environmental drivers
(Fig. A4).

High sums of strong interactions arise from temperature, precipitation, solar radiation, greffmin, emax and respcoeff (Fig.
A4a,b).
**Code and Data Availability**
LPJ-GUESS development is managed and the code maintained in a permanent repository at Lund University, Sweden.
Source code is made available on request. The model version presented in this paper is identified by the permanent revision
number r10207 in the code repository. There is no DOI associated with the code. Code to perform the sensitivity and

**Author contribution**

JO and FH conceived and designed the study and wrote a first draft. JO implemented the case studies, ran the experiments,
and analyzed the results. CH, AK and PA advised regarding running the LPJ-GUESS model. CH, AR and AK determined
the prior ranges for the parameters. All authors contributed to discussing and interpreting the results, and to the preparation
of the manuscript.

**Competing interests**

The authors declare that they have no conflict of interest.

**Acknowledgements**

555  We acknowledge funding from the Bavarian Ministry of Science and the Arts in the context of Bavarian Climate Research

556  Network (bayklif). We thank the LPJ-GUESS developers for developing and maintaining the LPJ-GUESS model. We also

557  thank two anonymous reviewers for their valuable comments and feedback on an earlier version of the manuscript.

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

**Tables**
**Table 1: The model inputs investigated in the sensitivity analysis can be group in a) common parameters b) species-specific**
**parameters and c) drivers. The ranges for the parameters have been determined from experts and literature, default parameter**
**values that changed from Hickler et al. (2012) due to the reparameterization are explained in Appendix A1.2 .\* denotes an**
**averaging over sites.**

### a) Common Parameters

| Grouping | Parameter | Explanation | Unit | Default Value | Min. Value | Max. Value | Literature sources |
|---|---|---|---|---|---|---|---|
| Mortality / Management | distinterval | average return time for generic patch-destroying disturbances | year | 920 | 200 | 1000 | - |
| Nitrogen | nfix_a | First term in N fixation eqn | - | 0,102 | 0,102 | 0,367 | - |
| Nitrogen | nfix_b | Second term in N fixation eqn | - | 0,624 | 0,764 | 0,624 | Clivoloot et al., 1999 |
| Nitrogen | nrelocfac | Fraction of N retranslocated prior to leaf and root shedding | - | 0,5 | 0,1 | 0,8 | - |
| Photosynthesis/Light | lambda_max | Non-water-stressed ratio of internallar to ambient $CO_2$ pp | - | 0,8 | 0,6 | 0,8 | - |
| Structure/Phenology | repfrac | Fraction of NPP allocated to reproduction | - | 0,1 | 0,05 | 0,3 | Pappas et al., 2013 |
| Structure/Phenology | turnover_root | Rate of fine root turnover | 1/year | 0,7 | 0,65 | 0,75 | - |
| Structure/Phenology | crownarea_max | maximum crown area | mm^2 | 40 | 20 | 60 | - |
| Structure/Phenology | k_allom2 | height =allom2* -diameter^(allom3) | - | 60 | 30 | 80 | - |
| Structure/Phenology | k_rp | crown area = kallom1 -*height*(k_rp) | - | 1,6 | 1,3 | 1,6 | - |
| Water | emax | Maximum evapotranspiration rate | mm/day | 5 | 2 | 6 | Küstner 2000 |

### b) Species-specific Parameters

| Group | Parameter | Explanation | Unit | Pinus sylvestris Default Value | Min. Value | Max. Value | Picea abies Default Value | Min. Value | Max. Value | Fagus sylvatica Default Value | Min. Value | Max. Value | Literature sources |
|---|---|---|---|---|---|---|---|---|---|---|---|---|---|
| Establishment | parff_min | Min forest floor PAR for grass growth/tree estab | J/m^2/day | 2500000 | 1500000 | 3500000 | 1000000 | 750000 | 1600000 | 1000000 | 750000 | 1600000 | - |
| Establishment | gdd5min_est | Min GDD on 5 day C base for establishment | °C day | 500 | 250 | 700 | 350 | 300 | 700 | 1300 | 1050 | 1450 | - |
| Establishment | tcmin_est | Min 20-year coldest month mean temp for establishment | °C | -29 | -100 | -15 | -29 | -100 | -15 | -6,5 | -8 | -5 | - |
| Establishment | tcmax_est | Max 20-year coldest month mean temp for establishment | °C | 5,5 | -1,0 | 6 | 3 | -2 | 6 | 7 | 5 | 8 | - |
| Establishment | est_max | Max sapling establishment rate | 1/m^2/year | 0,2 | 0,1 | 0,25 | 0,1 | 0,05 | 0,2 | 0,2 | 0,05 | 0,25 | Schlbeski et al 2017 |
| Establishment | alphar | Shape parameter for recruitment-jux growth rate relationship | - | 10 | 4 | 15 | 4 | 2 | 5 | 2 | 2 | 5 | - |
| Mortality / Management | longevity | Expected longevity under lifetime non-stressed conditions (yr) | year | 500 | 300 | 900 | 300 | 200 | 1000 | 400 | 250 | 650 | - |
| Mortality / Management | fireresist | fire resistance | - | 0,4 | 0,05 | 0,7 | 0,1 | 0,05 | 0,8 | 0,1 | 0,05 | 0,8 | - |
| Mortality / Management | cultinterval | Time until trees are cut | - | 90 | 40 | 140 | 90 | 60 | 120 | 105 | 80 | 140 | - |
| Mortality / Management | greff_min | Threshold for growth suppression mortality | kgC/m^2/yr | 0,21 | 0,07 | 0,26 | 0,135 | 0,03 | 0,19 | 0,02 | 0,001 | 0,13 | Pappas et al 2013 |
| Mortality / Management | drought_toleran_ce | Implements drought-limited establishment plus water uptake, from 0; total to 1: not at all drought-limited | - | 0,25 | 0,1 | 0,4 | 0,48 | 0,2 | 0,65 | 0,39 | 0,2 | 0,49 | - |
| Mortality / Management | thinning_intensit_y | percentage of treshold crown/coverage that is kept after thinning | - | 0,9 | 0,45 | 1 | 0,9 | 0,5 | 1 | 0,9 | 0,55 | 1 | - |
| Photosynthesis/Light | respcoeff | Respiration coefficient | - | 1 | 0,8 | 2,2 | 1 | 0,8 | 2,2 | 1 | 0,5 | 1,5 | - |
| Photosynthesis/Light | pstemp_low | Approx lower range of temp optimum for photosynthesis | °C | 10 | 6,75 | 15 | 10 | 6,75 | 14 | 8 | 8 | 20 | Monnink et al. 2019 Pelkenik-Vermüalen et al. 2015 |
| Photosynthesis/Light | pstemp_high | Approx higher range of temp optimum for photosynthesis (deg C) | °C | 25 | 16 | 30 | 25 | 16 | 30 | 20 | 20 | 30 | Zhang et al 2014 |
| Structure/Phenology | cton_leaf_min | minimum leaf C:N ratio | - | 31,90 | 27,32 | 38,37 | 38,37 | 31,9 | 43,16 | 24,06 | 22,7 | 27,19 | - |
| Structure/Phenology | sla | Specific leaf area | m^2/kgC | 8,56 | 7,812 | 9,3 | 11,52 | 8,7 | 15,1 | 43,08 | 28,33 | 48,23 | Menuccioni, M. Borosi., L. 2001.; Pelkenik-Vermüalen et al 2015.; Xiao et al. 2006 |
| Structure/Phenology | turnover_sap | Rate of sapwood turnover | fraction/year | 0,085 | 0,05 | 0,1 | 0,065 | 0,04 | 0,09 | 0,085 | 0,05 | 0,1 | - |
| Structure/Phenology | k_latosa | Tree leaf to sapwood xs area ratio | - | 3000 | 1800 | 5200 | 4000 | 2500 | 7000 | 5000 | 2500 | 8000 | - |
| Structure/Phenology | k_allom2 | height =allom2* -diameter^(allom3) | - | 30 | 15 | 60 | Values from common Parameters | | | Values from common Parameters | | | - |
| Structure/Phenology | k_chillb | Coefficient in equation for budburst chilling time requirement | - | 100 | 80 | 800 | 100 | 80 | 800 | 600 | 250 | 800 | Pappas et al 2013; |
| Water | gmin | minimum canopy conductance not assoc with photosynthesis | mm/s | 0,3 | 0,22 | 0,38 | 0,3 | 0,22 | 0,38 | 0,5 | 0,42 | 0,58 | - |

### c) Drivers

| Group | Parameter | Explanation | Unit | Default Value | Min. Value | Max. Value | Literature sources |
|---|---|---|---|---|---|---|---|
| Environmental Drivers | insol | Mean deviations solar radiation from standard scenario RCP 4.5 per site | W/m^2 | RCP 4.5 | -63,9* | 65,2* | - |
| Environmental Drivers | temp | Mean deviations temperature from standard scenario RCP 4.5 per site | °C | RCP 4.5 | -5,40* | 5,82* | - |
| Environmental Drivers | prec | Mean deviations precipitation from standard scenario RCP 4.5 per site | mm/month | RCP 4.5 | -6,18* | 6,27* | - |
| Environmental Drivers | co2 | Mean deviations co2 from standard scenario RCP 4.5 per site | ppm | RCP 4.5 | -95,4 | 237 | - |
| Environmental Drivers | ndep | Mean deviations nitrogen deposition from standard scenario RCP 4.5 per site | g/mm^2/year | RCP 4.5 | 5,30E-07* | -4,22E-07* | - |

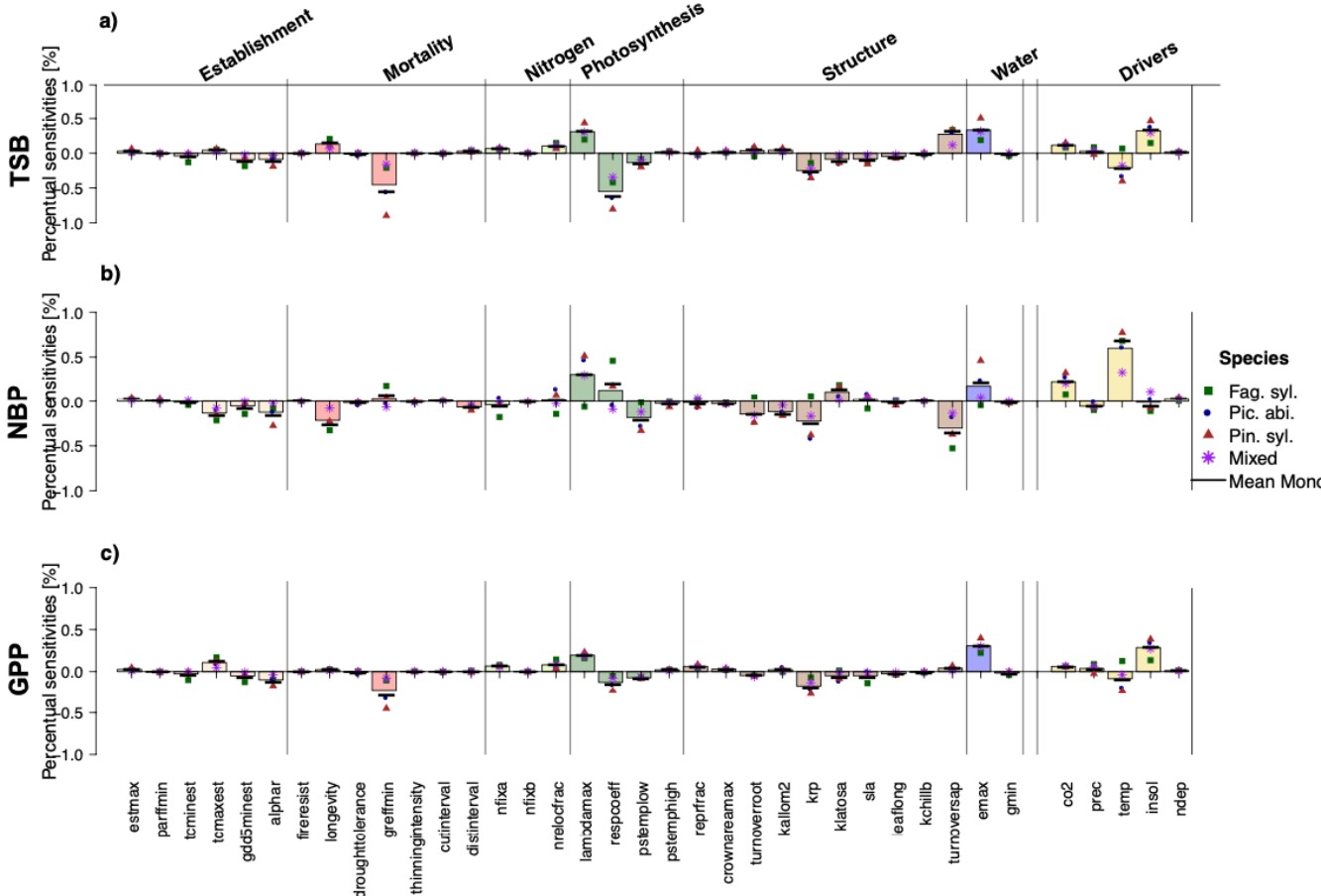


**Fig.1: Relative sensitivities (percent output change per percent parameter change) of the individual parameters and environmental
drivers regarding a) total standing biomass, b) net biome productivity and c) gross primary production. Sensitivities were not
substantially different between *Fag. syl.* (green squares), *Pic. abi.* (blue circles) and *Pin. syl.* (red triangles), but parameter
sensitivities were stronger for mono-specific stands than mixed stands (purple asterisks). The height of the bar reflects the mean
over mono and mixed stands. Positive values for points and bars indicate a positive and negative values a negative relationship
with the corresponding output.**

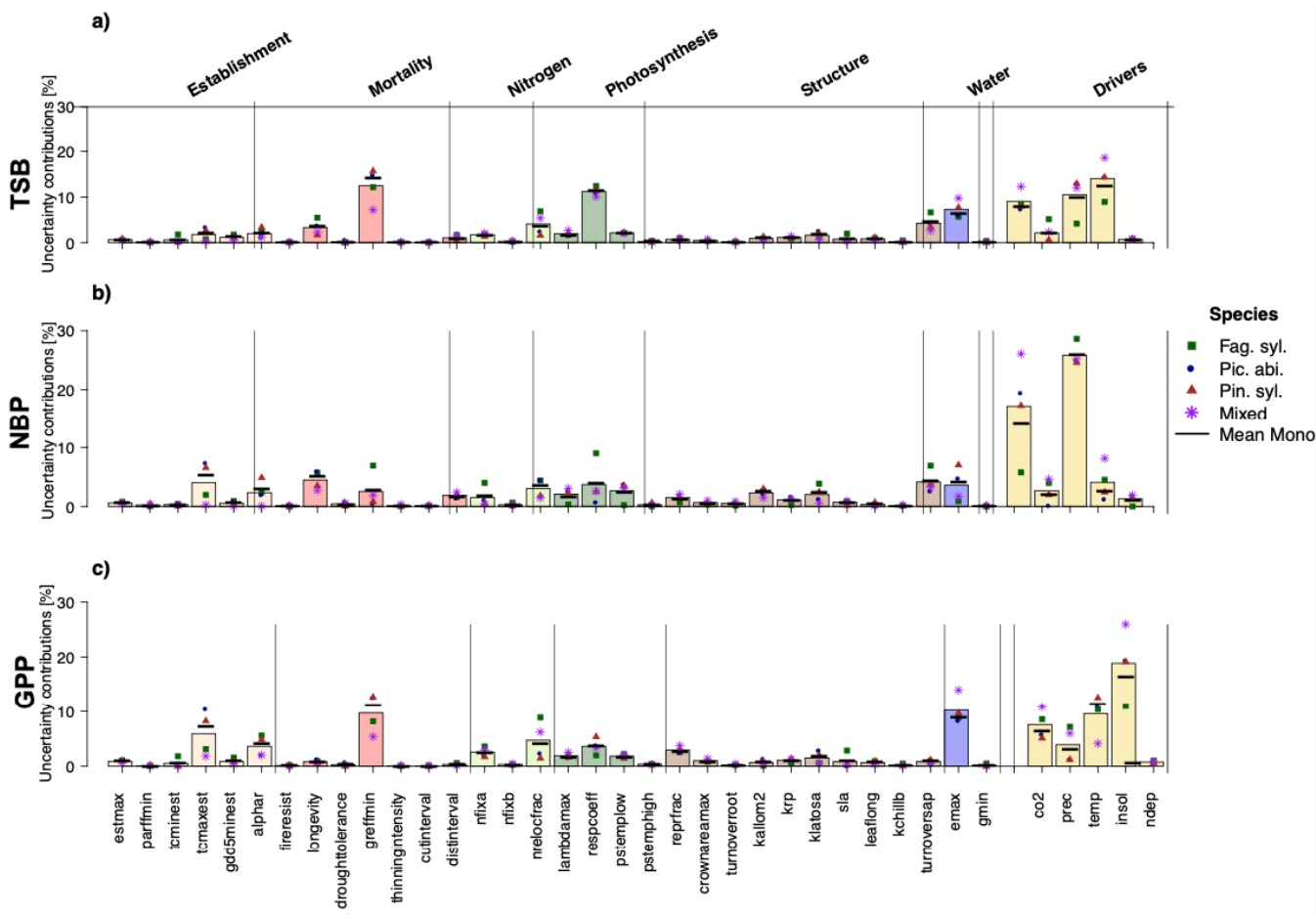


Fig. 2: Uncertainty contributions in percent of the individual parameters and environmental drivers regarding a) total standing biomass, b) net biome productivity and c) gross primary production showed no strong differences between *Fag. syl.* (green squares), *Pic. abi.* (blue circles) and *Pin. syl.* (red triangles) and were stronger for mono-specific stands than mixed stands (purple asterisks). The height of the bars reflects the mean over mono and mixed stands. Positive values for points and bars indicate a positive and negative values a negative relationship with the corresponding output.

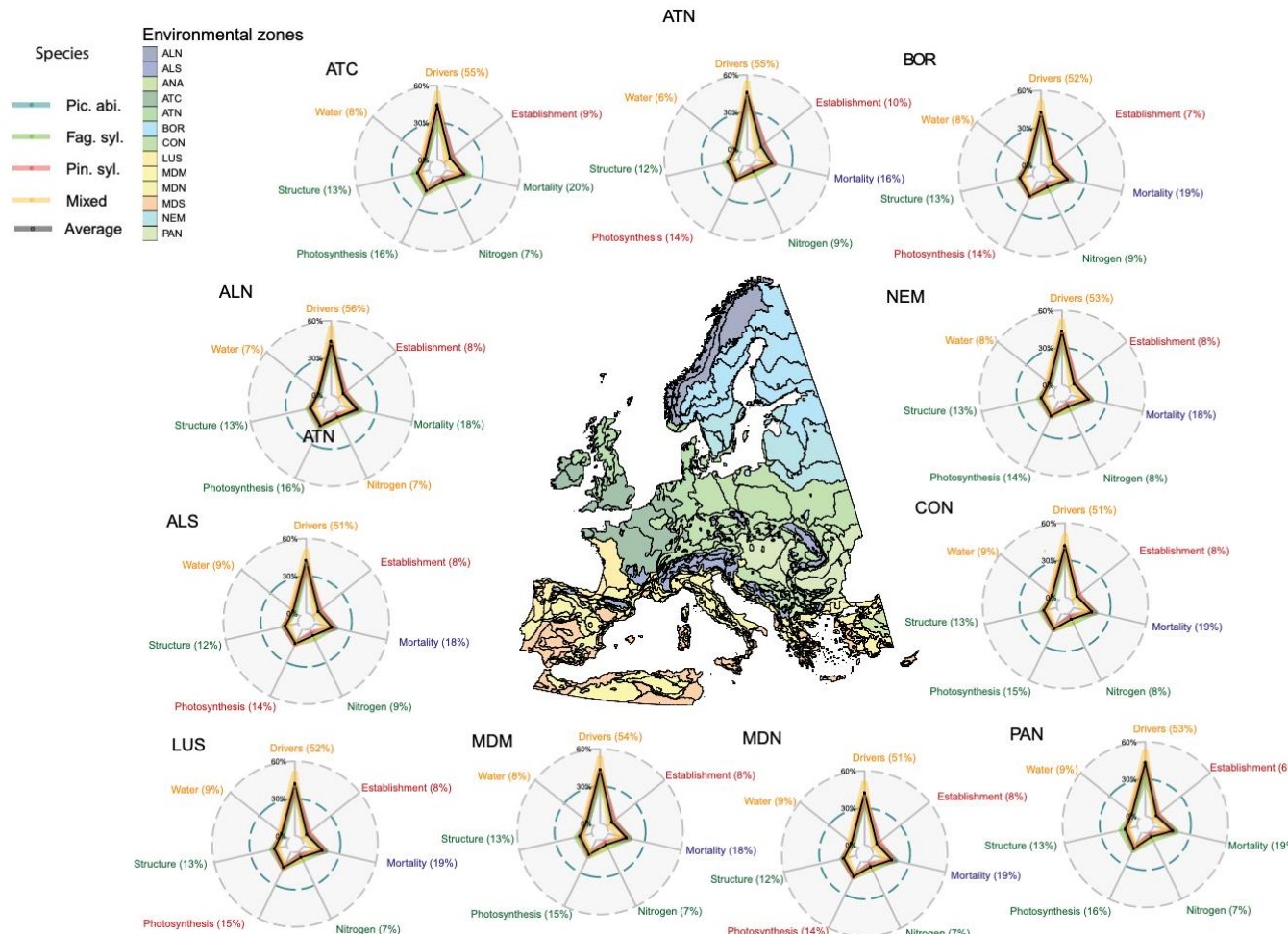

904

Fig. 3: The aggregated relative uncertainties of total standing biomass per environmental zone (with more than five sites) show a higher importance of drivers in the south than in the north. The environmental zones are from Metzger et al. (2005): ALN–Alpine North; ALS – Alpine South; ANA - Anatolian; ATC – Atlantic Central; ATN– Atlantic North; BOR–Boreal; CON–Continental; LUS – Lusitanian; MDM – Mediterranean Mountains; MDN – Mediterranean North; MDS – Mediterranean South; NEM – Nemoral; PAN – Pannonian. For each environmental zones the color and percentage value of the process label indicates which simulation setup (monospecific with corresponding species or mixed) has contributed most uncertainty and how much.


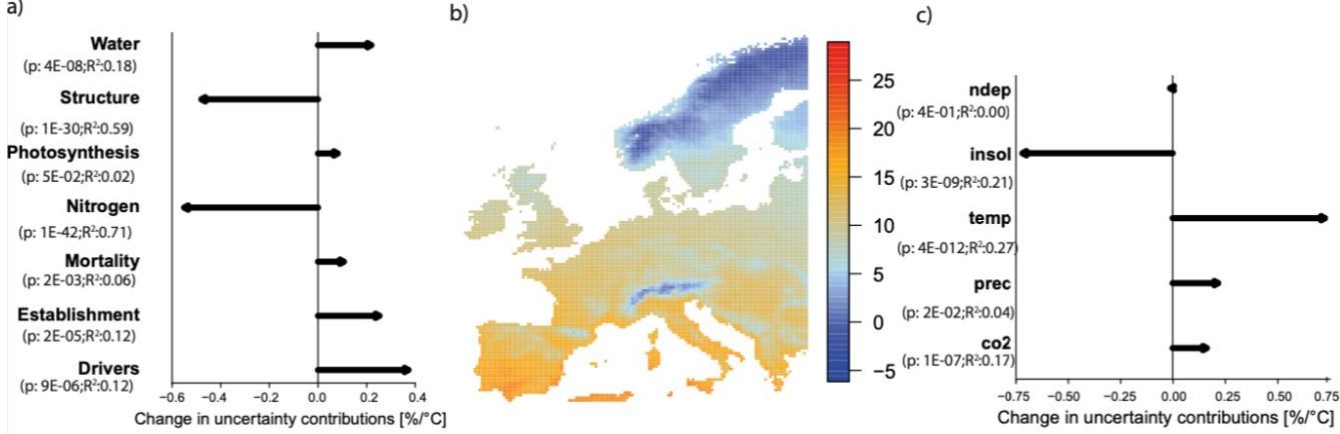


**Fig. 4: The uncertainty contributions to total standing biomass projections of parameters and environmental drivers change across**
**a mean annual temperature gradient across Europe from north to south (with p-values and $R^2$ for the processes/drivers). With**
**increasing temperature, the importance of drivers and establishment became higher for total standing biomass, while the**
**uncertainty contributions from nitrogen and structure declined (4a). The uncertainty contributions due to temperature increased**
**on the temperature gradient and the contributions from solar radiation decreased (4c).**

**Fig. 5: The induced uncertainty of environmental drivers, mortality- and photosynthesis-related parameters changed the most**
**depending on other parameters (Fig. 5a). Strong individual interactions between parameters and environmental drivers in**
**monospecific projections of total standing biomass were rare (Fig. 5b). If strong interactions occurred, these were mainly between**
**two environmental drivers or environmental drivers and parameters and only rarely between two parameters (Fig. 5b).**

**Tables Appendix A**
**Table *A*1: Differences in parametrization of Hickler et al. 2012 and our study for the investigated species (Fag. syl.,**
**Pic. Abi. and Pin. Syl)**

| Parameters | *Fag_syl* | | *Pic_abi* | | *Pin_syl* | |
|---|---|---|---|---|---|---|
| | Hickler et al. 2012 | Our study | Hickler et al. 2012 | Our study | Hickler et al. 2012 | Our study |
| drought_tolerance | 0.3 | 0.3 | 0.43 | 0.48 | 0.25 | 0.25 |
| fireresist | 0.1 | 0.1 | 0.1 | 0.1 | 0.2 | 0.4 |
| leaflong | 0.5 | 0.5 | 4 | 7 | 2 | 4 |
| turnover_leaf | 1 | 1 | 0.33 | 0.1429 | 0.5 | 0.25 |
| turnover_sap | 0.085 | 0.085 | 0.05 | 0.065 | 0.065 | 0.085 |
| est_max | 0.05 | 0.1 | 0.05 | 0.1 | 0.2 | 0.2 |
| alphar | 3 | 10 | 2 | 4 | 6 | 10 |
| parff_min | 1.250.000 | 1.000.000 | 1.250.000 | 1.000.000 | 2.500.000 | 2.500.000 |
| tcmin_surv (minimum 20-year coldest month mean temperature for survival) | -3.5 | -7.5 | -30 | -30 | -30 | -30 |
| tcmin_est (min. 20-year coldest month mean temperature for establishment) | -3.5 | -6.5 | -29 | -29 | -30 | -29 |
| tcmax_est (max. 20-year coldest month temperature for establishment) | 6 | 7 | -1.5 | 3 | -1 | 5.5 |
| twmin_est (minimum warmest month mean temperature for establishment) | 5 | -1000 | 5 | -1000 | 5 | 8 |
| k_chillb | 600 | 600 | 100 | 100 | 100 | 100 |

| sla | 43? | 43.08 | 11? | 11.52 | 8? | 8.56 |
|---|---|---|---|---|---|---|
| k_allom2 | 40 | 60 | 40 | 60 | 40 | 60 |
| wooddens | 200 | 293 | 200 | 185 | 200 | 211 |
| longevity | 500 | 400 | 500 | 300 | 500 | 500 |
| ga (aerodynamic conductance) | 0.04 | 0.04 | 0.14 | 0.14 | 0.14 | 0.14 |
| gdd5min_est | 1500 | 1300 | 600 | 350 | 500 | 500 |



**Figures Appendix A**

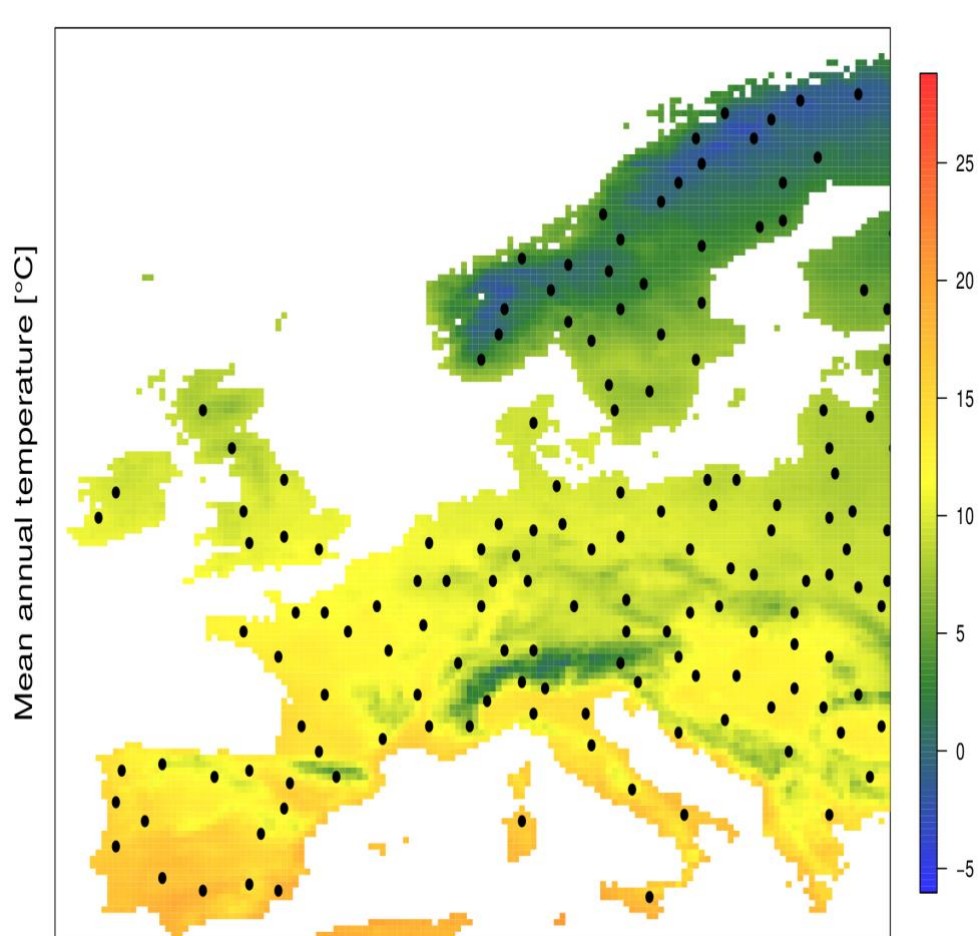


**Fig. A1: Our 200 sampled sites geographically and environmentally stratified over Europe cover the most important countries,**
**climate and temperature zones.**

**Parameterization as in Hickler et al. (2012)**

**Re-parametrization to fit to actual distribution**

*a) Picea abies*

*b) Picea abies*

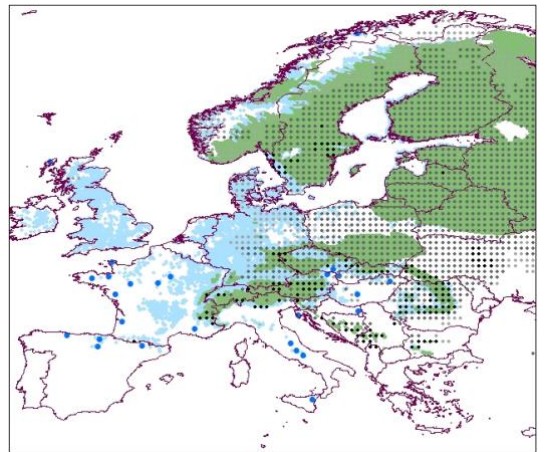
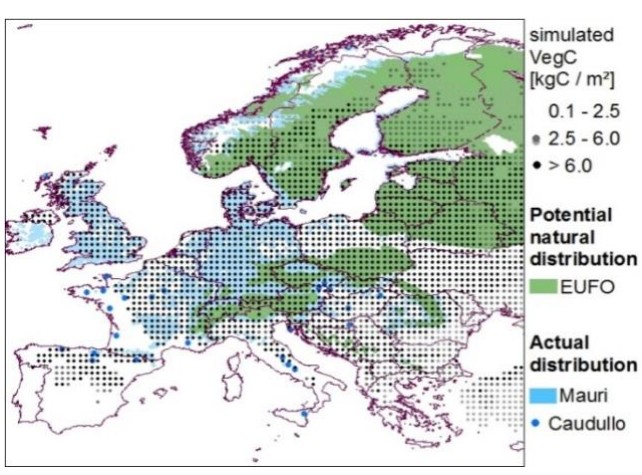

*c) Pinus sylvestris*

*d) Pinus sylvestris*

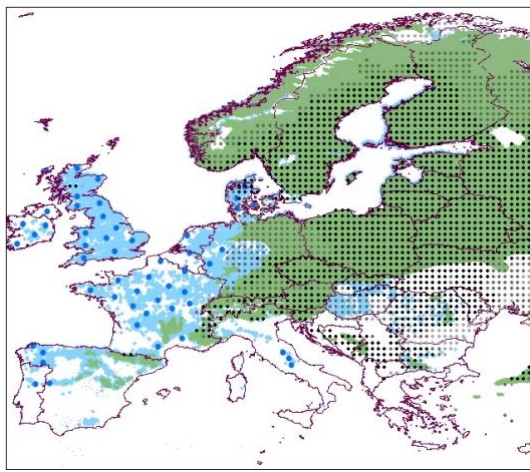
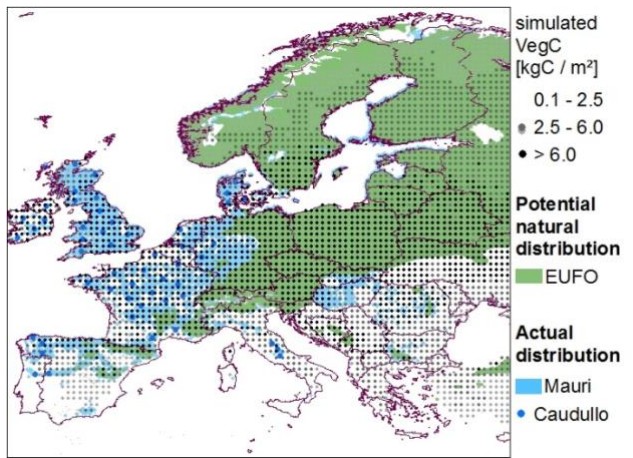

**Fig. A2: Simulated (black points), observed (blue) and natural distributions (green) of the adjusted parametrization (b, d) compared to applying the parametrization from Hickler et al., 2012 (a, c) for Picea abies and Pinus sylvestris. EUFO = EUFROGEN, 2008 and 2013, Mauri =(Mauri et al., 2017), Caudullo =(Caudullo, 2017). The simulations were run from 1600 to 2010 without management and without competition between species. The plotted biomasses were averages over the last 20 years.**

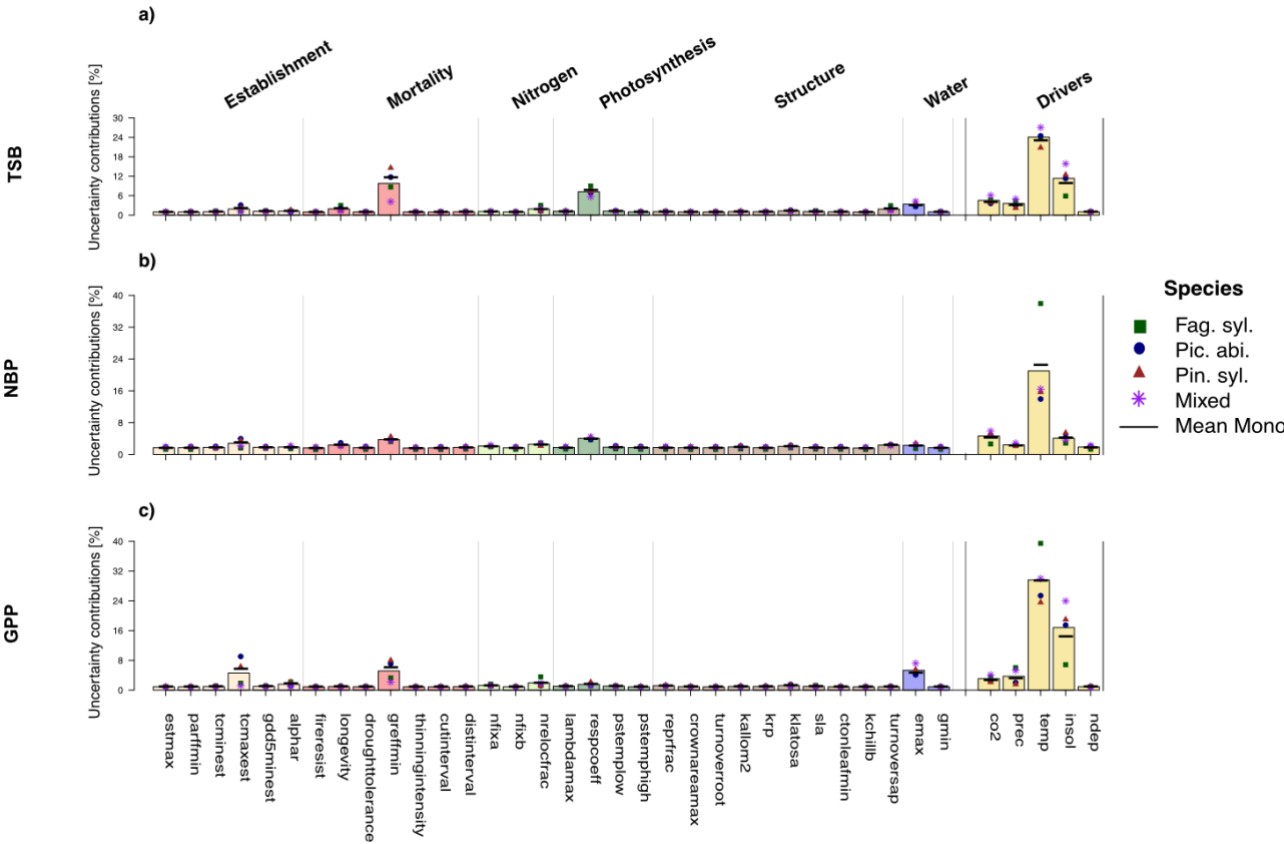


**Fig. A3: Results of the random forest uncertainty contributions. The uncertainties due to environmental drivers are higher than**
**the uncertainties due parameters compared to linear regression, but the ranking of parameters is similar to linear regression**
**results.**

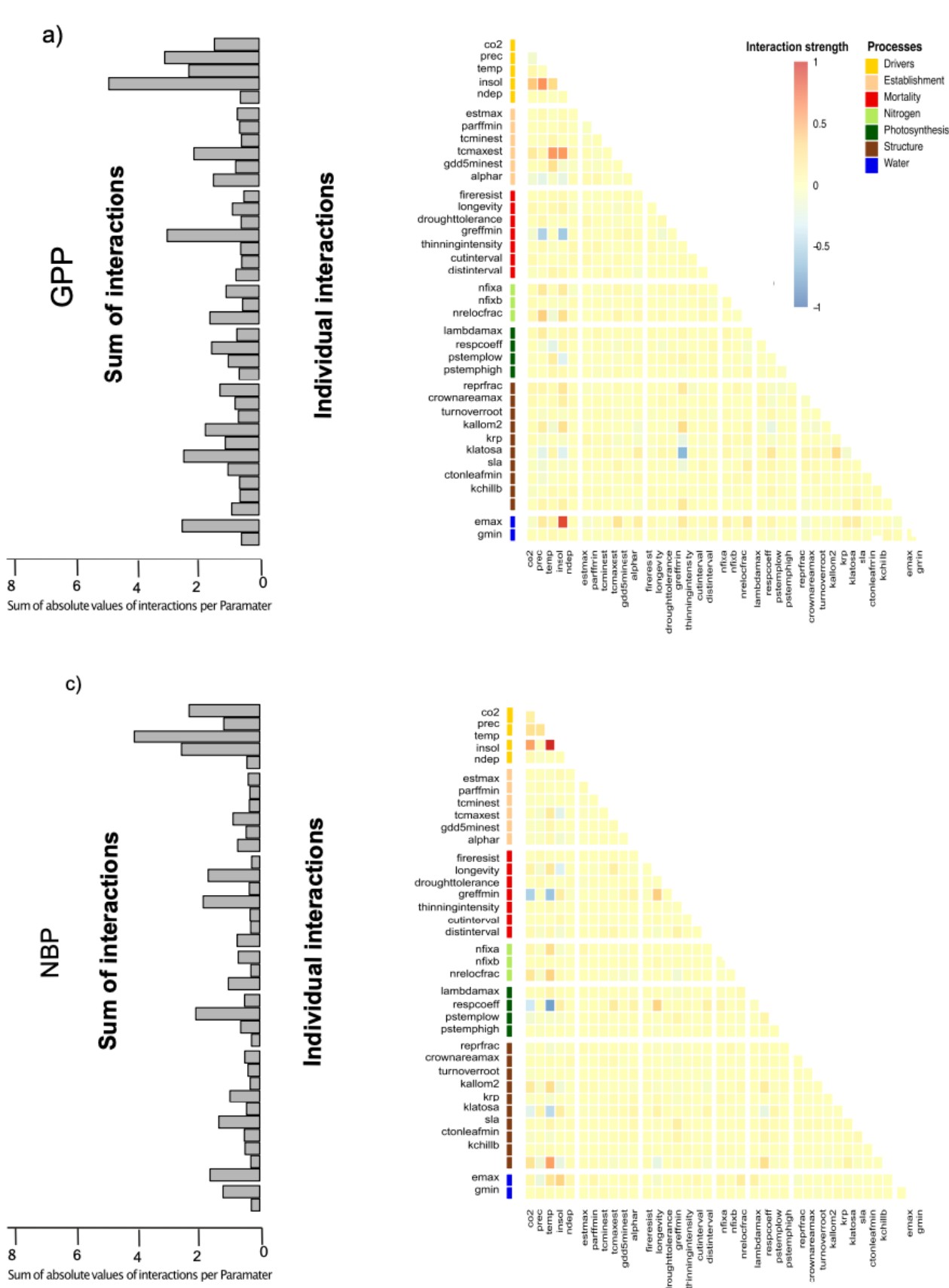

**Fig. A4: Interactions of uncertainty contributions of GPP and total standing biomass are similar to net biome productivity with**
**most interactions arising from environmental drivers.**