# Peer review of "Climate and parameter sensitivity and induced uncertainties in carbon stock projections for European forests (using LPJ-GUESS 3 4.0)"

_Geoscientific Model Development, 2021_

## Author Comment (AC1)

Dear editors of Geoscientific Model Development and dear reviewers,

we would like to thank all of you for your time and valuable feedback.

We have carefully considered all  comments of the reviewers and hope you will find that our changes to the manuscript, as well as the responses below, address all concerns that were raised. We want to highlight in particular the following changes:

- Reviewer 2 had concerns about the focus of our study. We changed it to focus more on the modeling perspective and the new processes in LPJ-GUESS.
- Both reviewers wanted us to add a paragraph regarding future model development, which we have done.
- Both reviewers mentioned that Fig. 3 is not clear enough. We changed the figure layout and hope that this makes the figure easier to grasp.
- Moreover, we carefully revised the discussion and took care to separate which claims are directly supported by the study, and which are based on our interpretation of the data.

Please find below our detailed response.

**Reviewer 1**

The manuscript by Oberpriller et al. with the title "Climate and parameter sensitivity and induced uncertainties in carbon stock projections for European forests (using LPJ-GUESS 4.0)" studies the sensitivities and uncertainties of a state-of-the-art dynamic vegetation model, LPJ-GUESS, with respect to its parameters and climatic drivers. Although I wished to see more species (afterall the title says "European forests") simulated, the study already has a quite comprehensive analysis protocol and it reports useful results as is. As the authors argue, the model has changed substantially since its last UA/SA and it is being used widely without these sensitivities and uncertainties being assessed properly. I would have also liked to see more links back to the development efforts, e.g. were (will) the new changes to the model structures (be) justified at the expense of introduced uncertainties? Maybe such aspects of the discussion could be fortified. But overall, I think this is a highly relevant study for the community, it is clearly written and I recommend its publication. I only have minor suggestions (numbers refer to lines):

Author's Response: Thank you for your positive evaluation and your useful comments!

24: Regarding "predictive uncertainty increases with temperature" I wonder if this is sort of an artefact of choosing relatively northern species. Could it help readers if simulated species are explicitly mentioned in the abstract already? It may also help with the connection to the "stress-gradient hypothesis" (i.e. how you define "harsh environment" may change from species to species).

Author's Response: This is true, we have added the investigated species in the abstract, as this is a central choice in our manuscript and the reader should know this as soon as possible.

25: dominates what?

Author's Response: This referred to carbon projections, corrected.

31: This sentence reads a bit unclear, maybe just focus on the latter part (differing in future carbon projections, although note that models also differ for the observable past; Bastos et al., 2020 https://doi.org/10.1029/2019GB006393)

Author's Response: Thank you, we agree and now focus on the second part. We have also added the reference regarding different observable past. The text now reads:

"Projektions from different vegetation models, however, often disagree on important details, for example regarding the observable past (Bastos et al., 2020) or the future carbon uptake of forest ecosystems (Huntzinger et al., 2017; Krause et al., 2019)."

33-35: Could also include initial condition uncertainty (Dietze 2017 https://doi.org/10.1002/eap.1589)

Author's Response: Agreed, added and cited.

46-47: Although, isn't SA most effective when magnitudes of uncertainty & variability are known, i.e. if one varies something by 1%s, whereas its natural variability is by 10%s, SA could be misleading.

Author's Response: We agree that SA can be misleading when the goal is to compare the importance of the parameters for the predictive uncertainty, and that differences occur in particular when the magnitude of uncertainty is different

between parameters. This is what we wanted to stress with these comments, and why we provide both SA and UA results.

On the other hand, when looking from the point of view of a model developer, an SA is often more informative than an UA, because a model developer wants to know how the model behaves, irrespective of what uncertainties were specified.

We therefore think that providing both UA and SA, where roughly UA = SA * uncertainty is the best practice. Typically, the UA results will be more relevant from the perspective of predictive uncertainty, while SA results will be more relevant to understand if the model behaves as expected.

We have summarized this in the text:

"It is important to note that uncertainties and sensitivities have different interpretations, and which of these two is more relevant strongly depends on the purpose. The calculated percental sensitivities can be interpreted as percentage change in the corresponding output, when changing a parameter value 1% in the prespecified range. The calculated uncertainties per parameter/driver can be interpreted as relative proportion of the overall uncertainty budget coming from environmental drivers and parameters. For scenario-analysis, e.g. comparing different cut intervals of forests, sensitivities provide a direct estimate of the model response, e.g. how much biomass changes when the cut interval is changed. For a comparison of different model forecasts, uncertainties are usually more relevant. If a reduction of uncertainty via a model-data comparison is the purpose, both measures are important, as parameters with high sensitivities can contribute more or less predictive uncertainty, depending on their input uncertainty."

We would love to expand on this in the paper, but there is already some discussion at the end of section 2.5, and we had the feeling that an extended discussion on the sense of UA and SA would be distracting here. No action taken so far, but would be happy to do so if the reviewer wishes.

68-75: Are these all for NPP? It would be nice to say which output variables' sensitivities and uncertainties they contribute to, if not NPP.

Author's Response: We have now added a more detailed description to the previous findings.

83: For other models, this may be a good citation here https://doi.org/10.5194/bg-17-2681-2020

Author's Response: Thanks, cited.

2.2. Simulation setup: Could authors add what they did for identifying soil types as well? I believe that's another input to the LPJ-GUESS model.

Author's Response: We used soil texture from Batjes (2005) and added this information to the text.

152-153: Are these (thinning intensity and cutting intervals) the only management parameters that arised from the new management module mentioned in lines 77-81?

Author's Response: No, there are more parameters, as for all other modules. As described, the choice of parameters to be included in the analysis was based on a systematic expert elicitation. In this process, no distinction was made between new and old parameters, i.e. experts were asked to identify the most relevant parameters for an SA overall.

We have added a comment that we do not use all parameters of the management module and refer to the original publication for more information.

"Additional mortality arises from forest management activities, determined by thinning intensity (percentage of all trees cut, *thinning_intensity*) and cutting intervals (*cut_interval*), which can be set for each species individually. For a more detailed description of the management module and the additional management parameters see Lindeskog et al. (2021)."

179-180: Could authors clarify what it means that they performed simulations as monospecific and mixed stands? I assume they ran LPJ-GUESS with all three species for all the 200 sites and let the model simulate whether the stand is monospecific or mixed, right? But the sentence reads as if they prescribed the model to simulate monospecific and mixed stands (i.e. ran LPJ-GUESS with one PFT only for monospecific sites).

Author's Response: We ran LPJ-GUESS on each of the 200 sites for each species (Pic. abi., Pin. syl, and Fag. syl.) individually and as a mixed stand. We have rewritten the sentence to make it more understandable.

"For these sites, we performed simulations for each of the three most common species in Europe (*Fagus sylvatica*, *Pinus sylvestris* and *Picea abies)* as monospecific stands and additionally all three species together as mixed stands."

212: Reads better if "respectively" comes before the numbers.

Author's Response: Thank you, done.

214: Now this note reads like authors pre-defined which of the sites are monospecific and which are mixed.

Author's Response: Yes, we agree that this sounded confusing. We rewrote our sentence in response to the previous and think that it is now clear what we have done.

217: "model output" == is this the average over all years?

Author's Response: Yes, we have added this to the sentence.

217-218: I think it may help clarify if authors state how many linear regressions they fitted in total in the end.

Author's Response: We added a sentence, which reads the following:

"We quantified sensitivity and uncertainty indices by running multiple linear regressions with the model output averaged over time as response, and parameters and drivers as well as their second order interactions as predictors. With 200 sites, each having three monospecific and one mixed stands setup, we overall ran 200x (3 +1) = 800 linear regressions. "

222: Random forest appendix section is A1.3, not A1.2

Author's Response: Thank you, done.

222-224: Just curious. Did the authors end up discarding any of the sites altogether? I.e. Were there any sites where none of the three species were sufficiently established?

Author's Response: No, at each site at least one species could establish enough such that the threshold did not force us to discard the site.

"We have chosen this threshold because smaller values indicate that the environment is not suitable for the species, however, for each site at least one species was able to establish."

Section 3.1 mean sensitivities: It might be worth noting some opposite behaviour in the sensitivity results as well, e.g. TSB and GPP are negatively sensitive to

temperature except for F. sylvatica. For NBP, something similar occurs for lambdamax, respcoeff, turnoverroot, krp and emax, amongst the important parameters.

Author's Response: We agree with the reviewer and accordingly added this to the text. The text reads now the following:

"Mixed stands were less sensitive to changes in parameters than mono-specific stands (Fig. 1). For monospecific simulations, species sometimes showed different magnitudes and even directions of sensitivities, especially *Fag. syl.* was more strongly affected by bioclimatic limits and *Pin. syl.* showed higher sensitivity to environmental drivers (temperature and solar radiation) than the other species. Moreover, TSB and GPP are negatively sensitive to temperature except for Fag. syl. For NBP, the direction of sensitivities changes between species for the non-water-stressed ratio of intercellular to ambient $CO_2$ *(lambdamax)*, the respiration coefficient (*respcoeff)*, the root turnover (*turnoverroot)*, an allometric constant (*krp)* and the maximum evapotranspiration rate (*emax)*."

Fig2: I think the alignment of the x-axis labels read better for Fig 2 (centered) than Fig 1. Also please consider sticking to the same order of parameters (x-axis) in both figures. Why is the mixed (*) symbol so much faded for NBP and TSB drivers? Finally, for sensitivity it makes sense (like authors say SA/UA have different interpretations) but isn't it a bit uncommon to visualize "negative contribution" to uncertainty? I mean, the caption says negative relationship not negative contribution, but still with the y-axis it reads confusing.

Author's Response: Thank you for the helpful comments on the figures. We now changed Fig. 2, such that it

- has the same ordering as Fig. 1 (aligned Fig. 1 labels as well)
- We only show the absolute value of these effects
- corrected b) and c) because overall this seems to be off somehow

Fig 3: Radar charts (or whatever you would call them) look almost identical to the eye, having more grids (inner circles) might aid the eye, or could plot an "average" polygon with a solid black line on each for relativity.

Author's Response: We have decided to plot the solid line with the average polygon.

Alternatively/in addition, for each process/driver the authors could add the highest percentage to the label, better yet if they use the color code (e.g. all "Drivers (X%)" would be orange for mixed).

Author's Response: We interpret this in the way that we should additionally report the highest percentage value per process and ENZ and indicate with the color to which simulation setup it belongs? If yes, we agree and we have done it.

I'm not sure how these would affect the readability but currently the information to ink ratio is rather low on this figure. Finally, "mortality" label seems to have strayed away from the BOR chart.

Author's Response: Adjusted the label.

296: Where did pH come from? This is the first and only time it appears in the manuscript text. It also appears unexpectedly on Fig4. Please clarify.

Author's Response: Thank you for spotting this issue. In a previous  version of the manuscript and simulations, we also included pH as a driver, but decided to remove it.

Fig 5: It could help if panel a bars were also filled with the panel b color scheme (orange, pink, red, light green etc.) for process groupings.

Author's Response: We agree and have done it.

310: I guess one of the interactions in the parentheses was supposed to be radiation-temperature.

Author's Response: Changed, thank you.

334-337: I think this is a very useful sentence in terms of guiding future development efforts. I wonder if authors could dedicate a short paragraph in the discussion (or fortify this one and the next) to summarize recent changes in the model structure and whether corresponding parameters stuck out in their SA/UA (it could also be important to state if they didn't show up). For example, nitrogen parameters showed up (and they were novel) but among them *nrelocfrac* contributed more. Management module was new but parameters didn't show up in the SA/UA. Do we learn anything from these?

Author's Response: We agree with the reviewer and have dedicated a paragraph in the discussion in the revised version to these issues. This greatly improved the importance of the manuscript and is useful for DGVM development. The text reads the following:

"**4.2. Associated uncertainties of previous changes in model structure and implications for future model development**

The management and the nitrogen cycling module are the most recent improvements of the LPJ-GUESS model (Smith et al., 2014; Lindeskog et al., 2021). Compared to previous sensitivity and uncertainty analysis, the high contributions of the nitrogen fixation to the predictive uncertainty of TSB and GPP (Fig. 2 a,c) are novel, though not surprising, as nitrogen is an important factor for the productivity of most temperate and boreal ecosystems (Vitousek and Howarth, 1991). The main reason why few earlier studies report those uncertainties is that vegetation models have only recently begun to integrate nitrogen cycling and limitation (e.g. B. Smith et al., 2014). The management module showed only small uncertainties, which could be due to the narrow parameter ranges for the cut interval and thinning intensity reflecting typical forest owners' choices. As forest owners usually try to maximize their profits (Johansson, 1986; but see Brazee and Amacher, 2000) and thus biomass production, low sensitivities of the management module are not surprising. A more suitable and important test case and application of the management module is a historical reconstruction of foliage projective cover data or similar outputs of the LPJ-GUESS model.

Our study helps to guide the model application, discussion of uncertainties and model development of LPJ-GUESS and other DGVMs. First, future model applications and model comparisons should focus on mortality as these processes contributes high uncertainties for carbon-related projections (see Fig. 1-3). Thereby, it should be investigated if these uncertainties stem from the intra-specific variability of the parameters itself (Bolnick et al., 2011), parameters are just not identifiable (see Marsili-Libelli et al., 2014), or if a model data comparison could reduce uncertainties in the parameters (e.g. Hartig et al., 2011). Using time series inventory data might help as it is informative for constraining mortality modules (Cailleret et al., 2020). Second, small sensitivities of establishment related parameters are surprising as we know that not all three investigated species can effortlessly establish across all of Europe, e.g. Fag. syl. can only establish on locations with no extreme drought and heat and no extreme winter frosts (Bolte et al., 2007). Thus, either we missed important parameters of this module, or the parametrization of the model needs to be updated. Third, when introducing new processes or coupling with other models (e.g. Forrest et al., 2020) calculating interactions helps to get a first impression where these new processes influence other model processes and potentially detect missing links. Moreover, future model applications can interpret their results with regard to the sensitivities in different factors (Saltelli et al., 2019) and discuss uncertainties and the causing factors, when used in policy advice (Laberge, 2013)."

337-338: I'm confused. On lines 71-72 authors said high sensitivities to water-related parameters were found: " Additionally, LPJ-GUESS showed high sensitivity to [...] water-related parameters (minimum canopy conductance not associated with photosynthesis, maximum daily transpiration, Pappas et al., 2013; Zaehle et al., 2005)." Please clarify.

Author's Response: Thank you, it is true that we have missed this. We corrected it in the revised version.

358-359: I thank the authors for explaining the potential cause of the negative effect, but while temperature affects TSB and GPP negatively, how does it affect NBP positively in the model?

Author's Response: We cannot say for sure, but note that, in contrast to TSB and GPP, the calculation of NBP also includes respiration and disturbances. So we speculate that respiration decreases more strongly with temperature than GPP (there is also a strong neg. interaction of the respiration coefficient and temperature for NBP, see Fig. A4). However, respiration itself depends not only on temperature but also on different factors like precipitation. A detailed discussion of the complex mechanisms behind this positive influence is out of scope for this manuscript, and in combination with the comments of the second reviewer we decided not to cover this topic in the revised manuscript.

360-361: Not just in magnitude but also in direction?

Author's Response: Thank you, we have added the direction to the revised manuscript.

367: Random forest results could be mentioned in the results section.

Author's Response: Done.

372-373: Does the finding "nitrogen-induced uncertainty decreases with increasing temperatures" correspond to the general statement of "limiting factors change across environmental conditions"? Or did the authors mean to cite a more specific ecological principle / hypothesis here?

Author's Response: We indeed meant the general principle that limiting factors change across environmental conditions. With the manuscript changes, this sentence however is not anymore in the manuscript.

373-377: Since the authors emphasize the stress-gradient hypothesis in the abstract and conclusion, I wonder if they can elaborate more and clarify the reasoning here to convince the reader. I had to read these sentences multiple times and it is still not clear to me how it follows from "decrease of uncertainty contributions of structure-related parameters on the temperature gradient" to the stress-gradient hypothesis which states where the physical environment is relatively benign (harsh) competitive (facilitative) interactions should be the dominant structuring forces. Water, mortality, establishment and photosynthesis parameters' uncertainty contributions also increase on the temperature gradient, which seem to indicate more competitive interactions to me.

Author's Response: We have now elaborated more on these questions in the manuscript and guided the reader towards our thought process. The entire paragraph now reads the following:

"Interestingly, our results of decreased uncertainty contributions of structure- related parameters and increased contributions of environmental drivers on the temperature gradient (Fig. 4) also seem in line with the stress-gradient hypothesis (Maestre et al., 2009), an empirically-observed pattern which states that in stressful environments, positive interactions should occur more often than in benign environments (e.g. Callaway, 2007). For the ecosystem that we consider, we interpret increasing temperature as increasing stress (e.g. Ruiz-Pérez and Vico, 2020), and structure as the best indicator for competitive interactions as the structure dictates resource allocation (e.g. bigger crown, but identical stem diameter leads to more photosynthesis; more sapwood to heartwood turnover requires less NPP). With this interpretation, one would conclude that under increasing stress, the importance of competition-related parameters decreases in the model, as expected from the stress-gradient hypothesis. We acknowledge that a fair amount of interpretation is needed to arrive at this conclusion, and we do not claim that this result lends evidence to the empirical discussion about the generality of the stress-gradient hypothesis, but we find it noteworthy that such a large-scale pattern emerges in the model from lower-level processes, without having been imposed (see also Levin, 1992)."

378: Unless it was an artefact of the analysis protocol (i.e. lines 341-342).

Author's Response: We have erased this topic from the manuscript, as we wanted to concentrate on the stress-gradient hypothesis and discuss this topic in more depth.

385: After conducting the analyses, (I know they say their results are robust to these choices but) were the authors happy with the uncertainty characterizations they initially came up with? I.e. do they still think these would be their best guess at this point or were there any parameters/drivers in particular that they wished they had varied/treated differently, for future studies?

Author's Response: We have no reason to think that our choices were bad, but also, we wouldn't know how to say that from the analysis we made. In effect, the results about UA are contingent on the uncertainties that are assumed, but there is no way to proof if these assumptions were good from the results of our study.

We realize, however, that our methods to quantify uncertainties were crude, in particular regarding the number of experts involved in the study. We do not think that this is a major limitation of the study, as we expect that experts should at least qualitatively agree about relative uncertainties, but including more experts and their opinion about plausible parameter ranges, for example, would have allowed us to

more exactly reflect the current view of the community about the uncertainty in each parameter.

We added a sentence about this into the text.

388: Roux & Buis et al. 2021 could be a good citation here
https://doi.org/10.1016/j.envsoft.2021.105046

Author's Response: Thank you, we have cited the paper now.

405: gradient-stress -> stress-gradient

Author's Response: Done

420 & 753: I guess I would avoid language such as "most important countries"

Author's Response: We fully agree and have changed the sentence.

424: Wasn't there any reparameterization in Smith et al., 2014 accordingly?

Author's Response: The parametrization in Smith et al., 2014 was for the global PFTs, but not for the species. There, the most recent parametrization was still Hickler et al., 2011.

425: "the productivity of trees in managed forests did not fit to the reported inventory data" where/when was this shown?

Author's Response: apologies for the missing reference. We base this statement on Fig. A2. Reference was added to the text.

445: Was the title supposed to say GPP and NBP? Likewise, check line 765 (Fig A4 caption)

Author's Response: Thank you for carefully reading this and finding our mistake.

**Reviewer 2**

The study analyzed sensitivities and uncertainties of the LPJ-GUESS 4.0 model for climate change simulations of 200 sites across Europe for three tree species in pure

and mixed-species stands. To this end, 11 general model parameters, 22 species-specific parameters and 5 environmental drivers were selected and varied within individually specified ranges. Sensitivities and uncertainties were calculated by multiple linear regressions for three model outputs relevant in the context of forests' role for carbon cycling (i.e., gross primary production, total standing biomass and net biome productivity). The results were analyzed for the whole of Europe, for environmental zones as well as along a temperature gradient.

Certainly, the results and insights of this study are relevant for simulation studies using LPJ-GUESS as well as other dynamic vegetation models and the manuscript fits well into Geoscientific Model Development. Yet, I have some remarks that should be addressed before publication.

**General/major comments**

Overall: The study provides valuable findings. However, the focus on overall ecological theory is not convincing and the comparison with empirical data very vague. I suggest changing the focus on the LPJ-GUESS model itself, its further development and the meaning of the results for recent and upcoming studies applying this model, particularly for publication in Geoscientific Model Development. Moreover, the manuscript would benefit from a significantly improved discussion of the findings, particularly in the context of LPJ-Guess and other dynamic vegetation models.

Author's Response: We agree that the focus of the paper should be mostly on the model and its implications for the DGVM community. Please see more detailed responses regarding this issue below.

Also, we would like to thank the reviewer for their careful revision of the manuscript, which has given us many good hints to improve the text!

1) The authors argue that one motivation for their study are substantial changes of the model structure, especially regarding nitrogen cycle and management modules, which have not been included in earlier SA/UA studies of LPJ-GUESS. While the process of nitrogen is mentioned and discussed, no insights are given regarding the management modules. An attentive reader can find out in Table 1 that two management-related parameters were considered in the analysis (cutinterval and thinning_intensity), which are however not related to any sensitivity/uncertainty according to Fig. 1&2. Please extend on the changes in model structure and include and discuss the findings in the discussion section.

Author's Response: We are unsure why the reviewer says these parameters are "not related" - the parameters certainly appear in Fig.1/2, but it is true that they are not particularly sensitive.

Upon finding that these parameters are not sensitive, one could of course argue post-hoc that the update of the SA compared to previous studies appears of lesser importance; however, this could have been known only after conducting the study, so testing the sensitivity (and possible interactions) of the these new parameters is necessary to know if the old results are still valid. In that sense, we see no problem with the motivation of our study.

It is true though that we had not put much emphasis on the sensitivity to management so far. We now elaborate on the changes in the management module of the model and included a section in the discussion about it.

2) Drivers vs parameters: It is not at all surprising that the environmental drivers contributed most uncertainty and had the highest sum of interactions in a climate-sensitive dynamic vegetation model as LPJ-GUESS. These drivers are inputs to all important processes in the model (i.e., primary production/growth, plant biogeography, soil hydrology, C exchange, etc.). Moreover, the variation ranges deduced from the different climate change scenarios are considerable. Hence, all the climate change simulation studies with LPJ-GUESS build on the sensitivity of the model to the climatic drivers. I can see that the added value of this study is that the uncertainty contributions can be attributed to the individual climatic drivers and analyzed across a temperature gradient. Yet, I think the authors should make really clear that parameters and drivers have different roles in process-based models. At least to me, it appears a bit as if you are comparing apples and oranges. In view of the potential insights for the LPF-GUESS modelling community, a separate analysis of both would show patterns much better (right now, 'drivers' mask all other changes in parameters in Fig. 3, which is not really informative).

Author's Response: We fully agree with the reviewer that drivers and parameters are conceptually different - this is why we refer to them as drivers and parameters throughout the manuscript, and never call temperature, for example, a model parameter. We have added a few sentences in the discussion to clarify this:

"As the model is sensitive to parameters and environmental drivers, and because these influence each other, we treated them in a combined sensitivity and uncertainty analysis (Saltelli et al., 2019), however, when interpreting it should be kept in mind that the one group relates to uncertainties in the model, while the other is external, so the two are conceptually very different."

That being said, from the point of view of a sensitivity or uncertainty analysis, both are factors that the model is sensitive to, or that can contribute to uncertainty, and that should therefore be included in a SA / UA. The need to include as many factors as possible to produce valid SAs/UAs is explicitly highlighted in:

Saltelli, Andrea, et al. "Why so many published sensitivity analyses are false: A systematic review of sensitivity analysis practices." *Environmental modelling & software* 114 (2019): 29-39.

Moreover, as we show, the two factor classes (Parameter, Drivers) influence each other, in the sense that the climate can change the sensitivity of the parameters. In such a situation, it is ill-advised to calculate sensitivities separately, as this could lead to over or underestimation of average (global) sensitivities. For this reason, Saltelli et al. stress in chapter "5.4. Recommendations for best practice"

*Both uncertainty and sensitivity analysis should be based on a global exploration of the space of input factors, be it using an experimental design, Monte Carlo or other ad-hoc designs.*

We would therefore argue that we follow the best advice in the literature, and that it would be counterproductive to make any changes to the methods.

3) I find it surprising that there are hardly any differences of the relative uncertainties across the environmental zones in Fig. 3. For example, the different species do not seem to show species-specific uncertainties to water-related parameters across space. I would also expect that not all species are able to grow in all environmental zones, which should somehow become visible at range limits. In a subsequent analysis, the authors focused on the uncertainty contributions across a mean annual temperature gradient (Fig. 4). In line with Fig. 3, the changes in uncertainty contributions are rather small (e.g., approx. 11% for temp between a 5° and a 20° site; e.g. Southern Sweden vs. Southern Spain). I wonder, whether these changes are statistically significant. Could you please provide some details (e.g., plots of the linear regressions including simulated TSB and $R^2$). Given these results, I got the impression that the authors oversold the results in this regard (L23-27; L371-380).

Author's Response: First, regarding the concern that not all species should be able to grow in all environmental zones: we are not sure about the entire environmental zone, but it is indeed so that some species were not able to grow on some plots. If

that was so, the species was excluded from the analysis for the given plot. This was noted in the methods section:

"To calculate mean sensitivities/uncertainties for each species, we averaged site-specific sensitivities over all sites with an average annual biomass production greater than 2 tC/ha. We have chosen this threshold because smaller values indicate that the environment is not suitable for the species, however, for each site at least one species was able to establish."

Moreover, note that we also varied the bioclimatic limits in the SA, meaning that boundaries for where species can grow are likely softer than if there would be a fixed parameter.

Regarding the concern of the reviewer that changes across the environmental zones are small: we would argue it is somewhat subjective if a 10% change is small or not. As the reviewer notes in point 2, LPJ-GUESS is climate sensitive, so it would be very surprising if temperature would affect model results in Scandinavia, but not in Spain. Note that what we display here are changes in percentage points in uncertainty contributions.

Moreover, we also vary the bioclimatic limits of each species and thus species basically have at least some parameter combinations for which they then can establish. Putting these two things together, the magnitude of differences in relative contributions across environmental zones do not seem surprising to us.

As requested by the reviewer, we now also report summaries of the fitted regressions (p-values and $R^2$).

4) Discussion: The discussion section needs considerable improvement. Various topics are mentioned, but there is very little substance and added value to many of the raised points (e.g., often only one sentence mentions the importance of a process and refers to a study that found similar effects). A better selection of the critical points and an in-depth discussion of these issues would greatly improve the manuscript.

Author's Response: We agree with the reviewer that we have a lot of topics to discuss in the manuscript. We have now focused on the most relevant topics which we discuss in greater detail.

Moreover, the discussion should be better embedded in the existing body of literature, both regarding model-based and empirical/physiological studies, especially if the authors keep the comparison with empirical results as one of the

four main objectives of this study (L102-103). Thereby, please make clear whether the reference you refer to is a model-based study or a field study.

Author's Response: We have now carefully revised the discussion to make this more clear, in which context this study is important and how it connects to the current literature.

Also, please be careful with the wording, e.g., a positive or negative effect of a parameter or driver can be explained by the fact that a certain ecological effect (which has been found by empirical studies) is integrated in the model formulation. Such an effect can be 'in line with empirical studies' but it cannot prove an effect as LPJ-GUESS is just a model.

Author's Response: We apologize if our wording gave the impression to the reviewer that we think that we prove an ecological effect by finding it in LPJ-GUESS. This was never our interpretation.

What we intended to do when comparing LPJ-GUESS behavior to ecological hypothesis is to show that the model behaves similar to what is reported or at least conjectured based on field observations.

We hypothesize that the model should behave similar to such field observations, and if it does, we primarily interpret this as evidence for the fact that the model behavior is plausible. We have carefully revised the manuscript to only make claims that are supported by this study.

Please also discuss processes, which turned out to be related to low sensitivities/uncertainties according to your simulation setup (e.g., establishment, management). Currently the discussion only considers the processes that turned out to be important, but it lacks an explanation why these patterns occur. For instance, no effect was found for management. This definitely needs explanation. Or, only small effects were found for establishment. Can this be explained by the spin-up and what are the implications for other simulation setups?

Author's Response: This is a good point. We have now dedicated a paragraph to a more in depth discussion of the results regarding the newer modules and modules of low sensitivity in LPJ-GUESS.

"Second, small sensitivities of establishment related parameters are surprising as we know that not all three investigated species can effortlessly establish across all of Europe, e.g. Fag. syl. can only establish on locations with no extreme drought and heat and no extreme winter frosts (Bolte et al., 2007). Thus, either we missed important parameters of this module, or the parametrization of the model needs to be updated."

Given the motivation the authors outline in the introduction, it would be worth to cover the following points: How are your findings relevant for recent and upcoming studies using the LPJ-GUESS model? What does your results imply with regard to further model development efforts? How could the robustness and reliability of the model projections be increased?

Author's Response: We agree that model development is a motivation for our analysis, but it is by far not the only one. In general, we see at least 3 purposes for an SA / UA

- Model application (it is useful for other studies, e.g. to interpret differences, to know how sensitive the model is to particular factors)
- Discussion of uncertainties - for policy advice etc., it is useful to know how large uncertainties are, and what is causing them.
- Model checking / development.

Of those 3, we see model development possibly as the least important, not least because it is not clear what sensitivities say about model development. After all, even a structurally perfect model would be sensitive to inputs / parameters, and we would be hard-pressed to distinguish between a good / bad model based on sensitivities.

Sensitivities can guide model developers and empiricists towards the areas in which empirical data would be particularly useful for reducing predictive uncertainties of models or guide parameter selection for model calibration for a specific region.

We have slightly modified the discussion to convey these ideas.

"Our study helps to guide the model application, discussion of uncertainties and model development of LPJ-GUESS and other DGVMs. First, future model applications and model comparisons should focus on mortality as these processes contributes high uncertainties for carbon-related projections (see Fig. 1-3). Thereby, it should be investigated if these uncertainties stem from the intra-specific variability of the parameters itself (Bolnick et al., 2011), parameters are just not identifiable (see Marsili-Libelli et al., 2014), or if a model data comparison could reduce uncertainties in the parameters (e.g. Hartig et al., 2011). Using time series inventory data might help as it is informative for constraining mortality modules (Cailleret et al., 2020). ".... "Our study helps to guide the model application, discussion of uncertainties and model development of LPJ-GUESS and other DGVMs. First, future model applications and model comparisons should focus on mortality as these processes contributes high uncertainties for carbon-related projections (see Fig. 1-3). Thereby, it should be investigated if these uncertainties stem from the intra-specific variability of the parameters itself (Bolnick et al., 2011), parameters are just not identifiable (see Marsili-Libelli et al., 2014), or if a model data comparison could reduce uncertainties in the parameters (e.g. Hartig et al., 2011). Using time series inventory data might help as it is informative for constraining mortality modules (Cailleret et al., 2020). "

**Minor comments**

*Abstract*

L23-27: The conclusions are strongly focusing on the stress-gradient hypothesis, which was not the main focus of this study. I am convinced that there are other, more general conclusions you could make. For example: How are your findings relevant for recent and upcoming studies using the LPJ-GUESS model? What are conclusions that also consider your findings regarding the model parameters? What does your results imply for further model development?

Author's Response: In the revised version, we have focused more on future model development and especially on the implications of our study on the parameters and processes. The text now reads:

" In conclusion, our study highlights the importance of environmental drivers not only as contributors to predictive uncertainty in their own right, but also as modifiers of sensitivities and thus uncertainties in other ecosystem processes. Reducing uncertainty in mortality related processes and accounting for environmental influence on processes should therefore be a focus in further model development."

*Introduction*

L58: please provide a reference for local sensitivity/uncertainty analysis

Author's Response: Done.

L83: no comma after 'model'

Author's Response: Deleted.

L86-88: difficult to follow as you introduce another aspect (ecological principles). Please split the sentence and explain better how the aspect of ecological principles can be evaluated by SA/UA and why this is interesting (and give references)

Author's Response: We agree that this was hard to read and have split the sentence and motivated the use of SA for ecological hypothesis testing in more detail.

About references for why it is interesting to compare the results of an SA/UA to ecological principles: we have no specific reference to SA / UA, but of course it is just the scientific method to say

1) if the model is correct
2) and model input e.g. temperature, is varied
3) then the sensitivity to temperature should be similar to what is observed in the field.

There are of course countless references on this (starting with Popper), and this is basically the logic we apply here.

We hope the reformulation of the sentence clarified our thinking. The text now reads:

"When sensitivities or uncertainties of parameters belonging to a specific process increase on an environmental gradient, this indicates that the process itself becomes more important on the gradient (Saltelli, 2002). By comparing such changes to existing ecological hypotheses, we can test if model sensitivities and thus process descriptions are in line with ecological expectations."

L92: how did you make sure that the 200 sites you selected were forest sites? Else, please do not refer to 'European forests' here.

Author's Response: No, we did not control for present vegetation, just for the potential that a forest can grow according to LPJ-GUESS (PNV). Removed the word forest.

L95: which processes were not considered? Could be added in the section 63-75.

Author's Response: Thanks, we came up with these process names and parameter attributions to the parameters by ourselves and tried to cover all processes. What we can say and have added is that we did not consider parameters from soil hydrology in this study. We did this in the text now:

"whereas soil hydrology parameters were not identified as very sensitive in earlier studies. "

L95-99: long sentence, consider splitting;

Author's Response:  Done

 why do you run simulations for pure and mixed stands? Please explain and mention relevant references. Are mixtures of these three species common in Europe (or do they stand for different life-history strategies)?

Author's Response:  We now added an explanation that we wanted to compare potential differences between mono and mixed stands, which were identified based on literature. The text now reads:

"We simulated the most abundant tree species in Europe (*Fagus sylvatica, Pinus sylvestris* and *Picea abies*) individually and in mixed stands, as these species are suffering from climate change (e.g. Buras et al., 2018; Walentowski et al., 2017) and could benefit from mixed stands (e.g. Pretzsch et al., 2015)."

L98: 'and' not in italic

Author's Response: Done

L101: introduce the environmental zones before and give a reference to the classification system you applied

Author's Response: Done

*Methods*

L108: add space after '2014).'

Author's Response: Done

L117: please give model version in this section

Author's Response: Done.

L117: is fire relevant in this study. If yes, please explain the BLAZE model briefly.

Author's Response: We have described this now in the text.

"In this model version, fire is based on the BLAZE model (Rabin et al., 2017). Thereby annually burned area is generated based on fire weather and fuel continuity and distributed to monthly intervals based on climatology (Giglio et al., 2010). Tree mortality is then estimated by computing firelines based on weather and converted into height-dependent survival probabilities (see Haverd et al., 2014) depending on empirical biome specific parameters."

L119: are these key parameters according to a previous SA/UA or key parameters because they are first in the modeling routine?

Author's Response: Thank you, we added that these arose from expert elicitation.

"A first set of key parameters from our expert elicitation (see below) for **establishment** are the bioclimatic limits (i.e. minimum growing degree days (*gdd5min_est*), minimum 20-year coldest month (*tcmin_est*), maximum 20-year coldest month (*tcmax_est*) and minimum forest photoactive radiation at forest floor (*parff_min*)), which build the environmental envelope for establishment."

L122: please move 'at regular intervals (here: 1 year)' to after 'are established'

Author's Response: Done.

L123: add 'is' after 'floor'

Author's Response: Done.

L126: please add 'net primary productivity' before NPP

Author's Response: Done.

L137: is there missing something after 'for'?

Author's Response: "for" here refers to nitrogen and water in the next part of the sentence. For better readability, we added it.

L144: please remove the comma after 'competition'

Author's Response: Done.

L148: does population stand for cohort or all trees of the respective species on the forest patch?

Author's Response: We changed this now to cohort and added an explanation.

L150: how are these wildfires linked to climatic conditions? Which trees are killed by fires (i.e., I guess not all as there are additional patch-destroying disturbances)?

Author's Response: In the revised version we have described this in the first paragraph mentioning fire (see comment above).

L151: please give examples for the patch-destroying disturbances

Author's Response: In LPJ-GUESS this is a generic patch-destroying disturbance. This could be a windthrow or landslide. We added these examples in the text.

L168-173: please add some references to previous LPJ-GUESS publications introducing this process

Author's Response: Done

L177: please give a reference to the method applied and a reference to the climate data you used

Author's Response: We have now cited the package used for random stratified sampling and for the climate data.

L179: replace 'und' by 'and'

Author's Response: Done.

L189: please add a point before 'We'.

Author's Response: Done.

L189: What type of data did you use to derive $CO_2$ values for the transient and future simulation runs?

Author's Response: We think the reviewer might have missed that we write "atmospheric $CO_2$ concentration from Meinshausen et al. (2011) "

L196: which fraction of all parameters are covered by these 11 and 22 parameters?

Author's Response: We have counted all the parameters in the instruction file and came up with an estimate of about 33% and respective 20% of the available parameters in the ins file. There are additional parameters, which are hard coded.

L208-209: please add references for carbon cycling and forest owners.

Author's Response: Done.

L211-228: does your approach correspond to a global or local SA/UA? Please clarify. Are there other SA/UA that followed the same approach/method as you?

Author's Response:  We have added in the text that is a global SA/UA and added reference to applications for other dynamic models.

"We quantified sensitivity and uncertainty indices by running multiple linear regressions with the model output averaged over time as response, and parameters and drivers as well as their second order interactions as predictors. With 200 sites, each having three monospecific and one mixed stands setup, we overall ran 200x (3 +1) = 800 linear regressions. This analysis corresponds to a global SA/UA in the context of regression analysis and has been applied to other system models (e.g. Sobie, 2009)."

L217-236: well explained

Author's Response: Thanks.

L230: replace 'are' with 'is'

Author's Response: Done.

L230: very good explanation for sensitivities. Please add such an explanation for uncertainties too.

Author's Response: Thanks and done. The text now reads:

"It is important to note that uncertainties and sensitivities have different interpretations, and which of these two is more relevant strongly depends on the purpose. The calculated percental sensitivities can be interpreted as

percentage change in the corresponding output, when changing a parameter value 1% in the prespecified range. The calculated uncertainties per parameter/driver can be interpreted as relative proportion of the overall uncertainty budget coming from environmental drivers and parameters. For scenario-analysis, e.g. comparing different cut intervals of forests, sensitivities provide a direct estimate of the model response, e.g. how much biomass changes when the cut interval is changed. For a comparison of different model forecasts, uncertainties are usually more relevant. If a reduction of uncertainty via a model-data comparison is the purpose, both measures are important, as parameters with high sensitivities can contribute more or less predictive uncertainty, depending on their input uncertainty."

*Results*

General comments:

- It is helpful, if you mention the process groups when referring to individual parameters. Sometimes you do this, but not throughout.

Author's Response: Done

- It would also be helpful to mention process groups that were not important too (e.g., establishment and nitrogen for 3.1)

Author's Response: We agree with the reviewer and added these for Fig. 1-2,5.

L250-252: The sensitivities of the three species are quite different from each other for some parameters (also different directions possible; not only for bioclimatic limits and environmental drivers). Please be more precise here.

Author's Response: We have now attributed that there were different magnitudes and even directions for the different species.

L263: please give numbers for the 'substantial' nitrogen-related uncertainty.

Author's Response: Done.

L270-271/284-286/L300-304: this should be part of the Method section. Please add the reference to Metzger et al. in the text.

Author's Response: We agree with the reviewer that this is usually placed in the methods section, however, we think a typical reader might miss the actual calculations if we place this somewhere in the methods section and the results are more clear, if we explain this here.

Reference was added in the introduction already.

L273: replace 'tree' by 'three'

Author's Response: Done.

L274: what does 'on average' mean? Averaged across three individual species and one mixture (if so, would it make sense to give the same weight to the individual-species simulations as to the mixture?)? Please clarify.

Author's Response: Yes this was averaged over the three monospecific and one mixed simulation. The weights in general should reflect on what you are interested in. If you are interested in absolute contributions of uncertainty, giving different weights might be a good choice, but as we wanted to see relative changes we prefered equal weights. The text now reads:

" On average across all environmental zones, stands and species about 45% of the uncertainty was due to environmental drivers, 15% due to mortality-, 14% due to photosynthesis-, 12% due to structure-, 7% due to water- and 7% due to nitrogen-related parameters (Fig. 3). "

L296: no comma before 'decreased'

Author's Response: Done.

L301: what do you mean by full dataset (in contrast to data used for Fig. 3 with at least 5 sites per environmental zone)? Please clarify.

Author's Response: We agree that "full dataset" might be distracting, thus we have erased it, without losing too much information. The sentence now reads:

"Interaction indices were calculated by averaging the interactions found in the linear regression over all sites and species."

L269: Does this analysis refer to TSB only? If so, please state clearly at the beginning and directly mention Appendix A1.4.

Author's Response: Yes this is true. Done.

L303: you summed the **absolute** individual interaction indices, right? What do you mean by 'other processes'? E.g., for $CO_2$, did you only took the sum of the interaction indices with parameters from other processes (i.e., without the other parameters from drivers, such as prec, temp, insol, ndep)? Please clarify.

Author's Response: We have now clarified that we have summed the absolute individual interaction indices. Moreover, we state now the influence on the other parameters/drivers, so it is clear that we summed up all. The text now reads:

*"Moreover, to investigate the overall influence on other parameters or drivers we summed the absolute individual interaction indices of each parameter with each other (Fig. 5a). "*

L307-309: similar to what?

Author's Response: "similar to each other". We have changed the text accordingly.

*Discussion*

General comment 1: Please make sure to be clear about which findings have been previously found for LPJ-GUESS and which for other DVMs.

Author's Response: We have carefully gone through the manuscript and precisely embedded our findings into the literature specifying which finding belongs to a) an empirical study, b) another DGVM or c) LPJ-GUESS.

General comment 2: The processes the authors discuss in a bit more detail seem to be picked rather randomly (e.g., in L344-351: some details about nitrogen, but nothing about water).

Author's Response: As the reviewer has mentioned earlier, we tried to discuss the new modules in more depth compared to the modules, which were already included in former SAs.

L322-326: The two sentences do state the same, right (deduced from two different figures)? This could be simplified as it seems confusing when reading through it. Consider skipping the sentence referring to Fig. 3, as this result can hardly be seen.

Author's Response: We have deleted the first sentence and included parts of it in the second sentence.

L328: consider deleting the interpretation here ('considering that ...'), as you give a summary of all results but an interpretation only for the very last one.

Author's Response: Done.

L330-351: Please extend a bit on your findings, otherwise these two sections are just a repetition of the introduction and your results. E.g., have you used similar parameter ranges as the previous LPJ-GUESS SA/UA studies? What do you think why have previous studies not found high sensitivies to water-related parameters (sites, species, parameter ranges, changes in model structure, ...)?

Author's Response: We have discussed less topics in more depth, also according to the major comment 4 so we removed some discussion points and concentrated on the important ones in this section.

L330-342: Do I understand correctly that for the mixed simulations, you did not change the values of a species-specific parameter simultaneously for all three species? But for the averaging, simulations for which the value of just one species has been changed for a specific parameter, where then averaged? Please clarify in the method section.

Author's Response: We did change the parameters of all species in mixed simulations simultaneously, but independent of each other. So for example there could be the case in the sampling design that we increased the maximum coldest month temperature for establishment for beech and spruce, but decreased at the same time for pine in one simulation. We have clarified this in the method section.

"Note, that for mixed simulations, for each simulation we individually drew parameter combinations for each species, i.e. the same parameter could be different for different species. In total, this means that 200 x (50.000 + 3 x 10.000) = 16 million LPJ-GUESS simulations were run."

L344-345: Please explain what is the difference to Petter et al. 2020.

Author's Response: Petter et al. also used different models and attributed most of the uncertainty to different models and not to climate. We included this information in the manuscript.

"We found that uncertainty contributions of environmental drivers were comparable to the uncertainty contributions of all parameters together (Figs. 2-5, see also Snell et al., 2018 for the FLMs model, but see Petter et al., 2020, who found that most uncertainty is induced by the choice of the forest model). "

L345: you did not mention nitrogen-related parameter as being important in the summary at the beginning of the discussion.

Author's Response: In the revised version it is mentioned in the summary.

L353: refer to the Figs via 'Figs. 2-5'

Author's Response: Done

L356-357: the positive CO2 effect could be explained by the assumed CO2 fertilization effect, which is integrated as an assumption in the LPJ-GUESS model. This does not necessarily mean that this is what happens 'in the real world', as various empirical studies questioned this effect (e.g. Körner and colleagues).

Author's Response:  A direct effect of Co2 fertilization is not modeled in LPJ-GUESS. The process of  photosynthesis and autotrophic respiration lead to the CO2-Response (see Hickler et al. 2008). These are modeled to our best knowledge reflecting the processes in nature. However, as phosphor cycling is not yet part of LPJ-GUESS, this could also be an explanation to the difference between simulated and observed Co2 response (see Fleischer et al. 2019 Nature Geoscience, which found less CO2 fertilization in CNP models compared to CN or C models). The text now reads:

"The positive effect of $CO_2$ could be explained by increased water-use efficiency and the $CO_2$ fertilization effect (also found for other DGVMs Keenan et al., 2011; Galbraith et al., 2010), which in LPJ-GUESS is an emerging property of the formulation of photosynthesis and respiration (see Hickler et al., 2008). However, empirical studies do not find such an effect (Körner, 2006), which could be link to the fact that LPJ-GUESS does not model phosphor cycling which could be the limiting nutrient (for a DVGM study see Fleischer et al., 2019). "

L361: replace 'is' by 'could be'.

Author's Response:  Done

I am not at all sure that this is true. Could your finding be related to the averaging effect across sites (e.g., Beech growing better across large areas of Europe due to northward spreads related to increasing temperature)? Please extend this point and explain your argument with empirical studies.

Author's Response: This could be also true, but we do not see how this contradicts our point: "higher resistance to increasing temperatures", which was also found in an empirical study, which we also cite.

L363-369: this paragraph is difficult to follow.

Author's Response: We have rewritten the paragraph.

L371-380: please consider major comment 4

Author's Response: Thanks, we focus on the stress-gradient hypothesis in more depth and deleted the other points and thus think we have incorporated major comment 4. The text now reads:

"Interestingly, our results of decreased uncertainty contributions of structure- related parameters and increased contributions of environmental drivers on the temperature gradient (Fig. 4) also seem in line with the stress-gradient hypothesis (Maestre et al., 2009), an empirically-observed pattern which states that in stressful environments, positive interactions should occur more often than in benign environments (e.g. Callaway, 2007). For the ecosystem that we consider, we interpret increasing temperature as increasing stress (e.g. Ruiz-Pérez and Vico, 2020), and structure as the best indicator for competitive interactions as the structure dictates resource

allocation (e.g. bigger crown, but identical stem diameter leads to more photosynthesis; more sapwood to heartwood turnover requires less NPP). With this interpretation, one would conclude that under increasing stress, the importance of competition-related parameters decreases in the model, as expected from the stress-gradient hypothesis. We acknowledge that a fair amount of interpretation is needed to arrive at this conclusion, and we do not claim that this result lends evidence to the empirical discussion about the generality of the stress-gradient hypothesis, but we find it noteworthy that such a large-scale pattern emerges in the model from lower-level processes, without having been imposed (see also Levin, 1992)."

L382-400: well written

Author's Response: Thanks

*Figures and Tables*

Table 1: Could you order the parameters by Group for better readability?

Author's Response: Done.

Fig. 1: The species result symbols are very small, please improve. The sensitivities of the three species are quite different from each other for some parameters (also different directions occur!). Please use the same y-axis for better comparability between the three outputs (also for Fig. 2).

Author's Response: We have enlarged the species symbols and used the same y-axis.

Fig. 3: I agree with reviewer 1.

Author's Response: Done.

Fig. 5: panel a x-axis label: change 'Param**a**ter' to 'parameter'; the shared y-axis labels for a) and b) is a bit confusing, particularly, since the axis labels are missing in a)

Author's Response: We have now colored the bars according to the processes and moved the labels more into the middle.

**Technical corrections**

Please add a comma: L73: after 'uncertainties'; L142: after 'period' and please add 'or' after 'PFT,'; L153: before 'which'; L158: after 'tissue'; L173: after 'coefficient'; L185: after 'data set'; L188: after 'data'; L227: after 'thereby'; L278: after 'zone'; L345: before 'especially'

Author's Response: Thank you, done

---

## Author Response (AR2)

Dear editor, dear reviewers,

Thank you for the positive evaluation of our manuscript. Please find below a point-by-point response to the technical comments. We hope this clarifies the remaining problems in the text, and will make the manuscript acceptable for publication.

On behalf of all authors
Johannes Oberpriller

**Reviewer 1:**

I thank the authors for addressing the comments by both reviewers. I recommend its publication with following minor comments (line numbers refer to the revised untracked manuscript):

L48: I would like to clarify that while I agree with the arguments authors make about UA and SA in general, I just think saying SA is agnostic of magnitudes of uncertainty could be a bit misleading. To me, knowledge about the magnitudes of uncertainty & variability is part of coming up with a proper prespecified range (L261) and expecting/exploring a certain behaviour from the model.

This feels like mostly semantics, but I would also say uncertainty propagation and sensitivity analysis are main tools of uncertainty analysis (L44), not the other way around (just like authors say in their response UA = SA * uncertainty). Overall I would revise this paragraph starting with L44 along the lines of:

"The two main tools to uncertainty analysis (UA) where the aim is to attribute uncertainty in model outputs to different inputs (drivers, parameters, and model structure) are sensitivity analysis (SA) and uncertainty propagation. The key difference between these two approaches is that uncertainty propagation considers the magnitude of uncertainty in the model inputs (e.g. parameters, typically determined via expert elicitations and previous studies) and translates them into uncertainty in model outputs, while SA translates a change in inputs into a change in outputs. Next, UA combines information from model sensitivity and input uncertainty to identify inputs with a high influence on model outputs, with the underlying idea that better constraining these will increase robustness and reliability of model projections."

I believe the rest of the text can remain unchanged but I leave the final decision to the authors, I don't think the overall results and messages of the study are affected by these definitions.

We understand what the reviewer means, but were not entirely happy with the formulation that was suggested. We have again re-formulated the paragraph, hoping to capture the spirit of this suggestion.

"The two main tools to understand how uncertainties in model inputs (drivers, parameters, and model structure) affect model outputs are sensitivity analysis (SA) and uncertainty analysis (UA) (Cariboni et al., 2007; Caswell, 2019; Saltelli, 2002; Saltelli et al., 2008). The key difference between these two methods is that in an UA, the central starting point is the quantification of uncertainty in the model inputs (e.g. parameters, typically determined via expert elicitations and previous studies (Matott et al., 2009)). This uncertainty is then propagated to the model outputs, and back-attributed to the different inputs. An SA, on the other hand, calculates how the model output changes per unit or percentual change of the respective input (Jørgensen and Bendoricchio, 2001). This calculation is primarily independent of the inputs' uncertainties, although local SAs can be affected by the reference point and global SAs by the range over which the sensitivity is calculated. Overall, however, both methods share the goal of identifying inputs with a high influence on model outputs, with the underlying idea that better constraining these will increase robustness and reliability of model projections (Balaman, 2019)."

As the reviewer stated that we are free to change or not, we assume this will be satisfactory.

L82: Could the authors provide citation(s) for this newly added sentence as well: "...whereas soil hydrology parameters were not identified as very sensitive in earlier studies."

Authors' Response: Done.

L350: I believe one of the interactions in the parentheses was supposed to be changed to radiation-temperature but it remains the same.

Authors' Response: Thank you, we indeed missed this and changed it now.

L441: Authors could consider citing Fisher et al. 2018 (a paper they're only citing in the first sentence currently) in this paragraph regarding the establishment and mortality discussion.

Authors' Response: Done.

L466: Authors could consider citing Dietze 2017b at the end of this sentence again.

Authors' Response: Done.

**Reviewer 2:**

I appreciate the substantial efforts the authors have invested into the revision of this manuscript. The manuscript has been greatly improved, particularly the discussion section. The study is now described in a concise way and the conclusions fit the objectives. The findings of this study, other modelling studies and empirical findings are now clearly separated, which strongly improves the discussion of the results.

All my comments have been addressed and I only have a few technical remarks:

Authors' Response: Thank you.

L147: Is there a noun missing after 'mechanical'?

Authors' Response: No, this was meant to read "mechanical (..) and functional balance". For better readability, we have added balance after mechanical as well.

L391: linked instead of link

Authors' Response: Done.

L438: 'would be' instead of 'is'

Authors' Response: Done.

L442: 'this process contributes' instead of 'these processes contributes'

Authors' Response: Done.

L884: 'grouped' instead of 'group'

Authors' Response: Done.

Colors of the processes (Drivers, Establishment, etc.) in Fig. 3: They are not consistent across the bioregions (e.g., mortality is either blue or green). Do these colors have a meaning? If yes, please add a short explanation. Else, you may consider to use the same colors as in Fig. 4?

Authors' Response:Yes, they have a meaning, described in the figure label, which was probably missed by the reviewer. It says there: "In the radar plots of each environmental zone, the color and percentage value of the process label indicates which simulation setup (monospecific with corresponding species or mixed) has contributed most uncertainty and how much. "